

# Hidden dualities in 1D quasiperiodic lattice models

Miguel Gonçalves[1], Bruno Amorim[2], Eduardo V. Castro[3,4] and Pedro Ribeiro[1,4]

**1** CeFEMA, Instituto Superior Técnico, Universidade de Lisboa,
Av. Rovisco Pais, 1049-001 Lisboa, Portugal
**2** Centro de Física das Universidades do Minho e Porto, University of Minho,
Campus of Gualtar, 4710-057, Braga, Portugal
**3** Centro de Física das Universidades do Minho e Porto, Departamento de Física e
Astronomia, Faculdade de Ciências, Universidade do Porto,
4169-007 Porto, Portugal
**4** Beijing Computational Science Research Center, Beijing 100193, China

## Abstract

We find that quasiperiodicity-induced transitions between extended and localized phases in generic 1D systems are associated with hidden dualities that generalize the well-known duality of the Aubry-André model. These spectral and eigenstate dualities are locally defined near the transition and can, in many cases, be explicitly constructed by considering relatively small commensurate approximants. The construction relies on auxiliary 2D Fermi surfaces obtained as functions of the phase-twisting boundary conditions and of the phase-shifting real-space structure. We show that, around the critical point of the limiting quasiperiodic system, the auxiliary Fermi surface of a high-enough-order approximant converges to a universal form. This allows us to devise a highly-accurate method to obtain mobility edges and duality transformations for generic 1D quasiperiodic systems through their commensurate approximants. To illustrate the power of this approach, we consider several previously studied systems, including generalized Aubry-André models and coupled Moiré chains. Our findings bring a new perspective to examine quasiperiodicity-induced extended-to-localized transitions in 1D, provide a working criterion for the appearance of mobility edges, and an explicit way to understand the properties of eigenstates close to and at the transition.


doi:10.21468/SciPostPhys.13.3.046

# 1   Introduction

Quasiperiodic systems (QPS) offer a rich playground to study localization without the presence of disorder. Like disorder, quasiperiodicity breaks translational invariance, but the nature of both phenomena is quite distinct - QPS are deterministic, while disordered systems are random and leads to drastically different localization properties. For systems with uncorrelated disorder, Anderson localization-delocalization transitions [1] are only possible in 3D. At lower dimensions, any degree of disorder localizes all eigenstates in the non-interacting limit [2,3] as long as no special symmetries are broken. For QPS, on the other hand, localization-delocalization transitions are possible even in 1D [4].

QPS host a plethora of exotic phenomena, manifested not only in nature of the possible localized/de-localized phases they can host , but also in nontrivial topological properties [5–7]. Interest in QPS has been recently renewed due to their experimental relevance in optical [8–15] and photonic lattices [5,7,16–18], cavity-polariton devices [19] and the promising recent developments in Moiré systems (see review in Ref. [20] and references therein). For the latter, effects of quasiperiodicity are often overlooked, but recent studies have shown that in some regimes these may have important consequences to localization and transport [21–26].

The first steps to understand quasiperiodic-induced localization were taken by Aubry and André who considered a nearest-neighbor tight-binding chain with on-site energies modulated by a sinusoid-the now celebrated Aubry-André model(AAM) [4]. Actually, this model was previously considered in a different context, to study the peculiar self similar energy spectrum of Bloch electrons under a uniform magnetic field [27, 28]. Aubry and André showed

that the AAM hosts a localization-delocalization transition arising simultaneously for all energies. This transition is a consequence of the real-space/momentum-space duality in the AAM's Hamiltonian. Generalizations of this model in 1D typically give rise to single-particle mobility edges [29–41] separating regions of the spectrum corresponding to localized and delocalized states. These may be highly non-trivial to determine and can only be analytically predicted for few fine-tuned models [18,30,33,34,42]. For higher dimensions, self-dual generalizations of the AAM show more exotic features around the self-dual points: intermediate phases, for which the wave function is delocalized both in real and momentum space and transport is diffusive [21,43]; and peculiar partially extended states [44]. The interacting version of the AAM was also recently studied. Here, many-body-localization for weak interactions and an intermediate regime with slow dynamics were found both theoretically [45–49] and experimentally [11,50,51].

Even though the study of non-interacting 1D QPS dates back to the 1980s, a simple framework to obtain phase diagrams for generic, non fine-tuned models and to understand the main ingredients behind localization-delocalization transitions is still lacking.

In this paper we argue that localization-delocalization [1] transitions in generic 1D QPS occur due to hidden dualities that are generalizations of the Aubry-André duality. With this insight, we developed a method to compute mobility edges and duality transformations through commensurate approximants (CA) of the target 1D QPS. To illustrate the power of the method, we obtained mobility edges and duality transformations for a number of models, including generalized Aubry-André models [30,31,34] and coupled Moiré chains, the paradigmatic example of a 1D Moiré system [23]. Our results established that dualities between localized and extended phases, up to now found for only a few fine-tuned models, are generic for 1D QPS and shows how they can be constructed.

The paper is organized as follows: in Sec. 3 we introduce a minimal set of concepts and give an summarized account of our work and main findings; in Sec. 3 we review in detail the duality in the Aubry-André model, paving the way for the discussion of more general hidden dualities. In Sec. 4, we provide details on the EAAM and LM models used throughout the paper. Sec. 5 details how the phase diagram of a QPS can be known by studying the generalized energy bands of CA. In Sec. 6 we provide a complete definition of our generalized duality transformation for CA and explain how to extract from it the duality transformation of the limiting QPS. We show that our duality transformation reduces to the thermodynamic-limit duality in a model for which the exact transformation is known [34]. We finally apply our duality transformation to models for which no duality symmetry was shown to exist so far.

## 2   Main Results

*Aubry-André duality* — We start by briefly reformulating the fundamentals of the Aubry-André duality that will be important to understand the generalized dualities introduced afterwords. The Hamiltonian for the AAM is given by [4]

$$H = t \sum_n \left( c_{n+1}^\dagger c_n + \text{h.c.} \right) + \sum_n V \cos(2\pi\tau n + \phi) c_n^\dagger c_n, \tag{1}$$

where the first term describes the hopping between nearest-neighbors, the second term is the quasiperiodic potential with strength $V$ and $\tau = a_1/a_2$ is the ratio between the nearest-neighbor distance, i.e. the lattice constant, $a_1$ and the potential wavelength $a_2$. From this point on, we measure lengths in units of $a_1$ by setting $a_1 = 1$, unless otherwise specified.

---

[1]Note that from this point on, we use the term "localization-delocalization transitions" to refer to transitions between non-critical extended and localized phases.

We consider a CA of an infinite system with irrational $\tau$ by taking rational approximations, $\tau_c = n_2/n_1 \simeq \tau$, where $n_1$ and $n_2$ are two co-prime integers [29,37]. The corresponding commensurate system is still infinite but becomes periodic with a unit cell of length $a_S = n_1 a_1 = n_2 a_2$ containing $n_1$ sites. Being periodic, the Hamiltonian can be block-diagonalized using Bloch's theorem. The Bloch Hamiltonian depends on the momenta $k \in \,]-\pi/n_1, \pi/n_1]$, and is periodic under $k \to k + 2\pi/n_1$. The block-diagonal sectors are labeled by the rescaled momentum $\kappa = n_1 k \in \,]-\pi, \pi]$, measured in units of $a_S^{-1}$.

We also define the rescaled shift variable $\varphi = n_1 \phi$. As we will see below, the energy bands are also periodic in $\varphi$ with a period $\Delta\varphi = 2\pi$ (or $\Delta\phi = 2\pi/n_1$). Importantly, the variables $\varphi$ and $\kappa$ are dual for any CA of the AAM and the extended and localized phases, as well as the critical point, can be inferred by studying the dependence of the energy bands on these variables. To do so, we define the generalized energy bands $E_n(\varphi, \kappa)$, with $n = 0, \cdots, n_1 - 1$. The generalized Fermi surfaces (FS) in the $(\varphi, \kappa)$ plane are defined by the constant-energy curves $E_n(\varphi, \kappa) = E$ .

The Aubry-André duality refers to a mapping between localized and extended wave functions [4]. In the potential-energy, $V - E$, phase diagram, points $P = (V, E)$ and $P' = (V', E')$ are dual if $(V', E') = (4/V, 2E/V)$, where here and in the following discussion, we set $t = 1$. Exactly at the critical point, $|V| = 2$, the wave function is equal to its Aubry-André dual. This point is therefore commonly referred to as a self-dual point. For a CA of the AAM, this duality still holds, but only under a suitable interchange of $\varphi$ and $\kappa$. Therefore extending AAM duality to commensurate structures naturally suggests a duality between the phases $\varphi$ and $\kappa$. We will prove all these claims in detail bellow.

In Fig. 1 we illustrate the duality of the FS in the $(\varphi, \kappa)$ plane for very simple CA of the AAM (with one- and three-site unit cells). At dual points the FS are the same upon interchanging $\varphi$ and $\kappa$. Moreover, as the unit cell of the CA is increased, the dependence of the FS on $\varphi$ ($\kappa$) decreases with respect to the dependence on $\kappa$ ($\varphi$) in the extended (localized) phase of the limiting QPS [compare Figs. 1(c,d)]. Exactly at the self-dual point, the FS are invariant under interchanging $\varphi$ and $\kappa$ [Figs. 1(e,f)]. The $\varphi \leftrightarrow \kappa$ duality holds irrespectively of the chosen CA.

*General dualities* — A brief overview of our findings and their application is summarized in Fig. 2. We consider two simple systems, an extended Aubry-André model (EAAM) and a ladder model (LM) consisting of two coupled incommensurate chains. For these two classes of models, $\tau$ is either the ratio between the lattice constant, $a_1$, and the QP potential's wavelength for EAAM, $a_2$; or the ratio between the two lattice constants for LM, $a_1$ and $a_2$: $\tau = a_1/a_2$. Despite the simplicity of these examples, we argue that the same basic mechanism extends to more complex 1D systems, including the so-called mosaic-models [52, 53] for which self-duality is believed to be absent.

Examples of CA for EAAM and LM are shown pictorially as rings containing a single unit cell in Figs. 2(a,b). The spectrum for a large enough unit cell can be seen in Fig. 2(c) and 2(d), respectively for LM and EAAM. We also compute the inverse participation ratio (IPR, see definition in Sec. 4), that distinguishes the extended (bluish) and localized (reddish) phases.

For a given CA we can again define the rescaled momentum $\kappa = n_1 k \in \,]-\pi, \pi]$. The Bloch Hamiltonian is equivalent to a single unit cell of the CA forming a ring with periodic boundary conditions that is threaded by a flux $\kappa$, as depicted in Figs. 2(a,b). $\kappa$ is also commonly mentioned as a phase twist. Besides the periodicity in Bloch momentum, the quasiperiodic models possess an additional periodicity related to the phase of the QP potential in EAAM and the stacking between the two chains in the LM. Both cases can be seen as a slide, or shift, degree of freedom between the potential and the lattice (in EAAM) of the two lattices (in LM) as depicted respectively in panels b) and a) of Fig. 2 . The crucial point is that, for a given CA, for certain shifts smaller than the CA unit cell, the system is left invariant (up to a relabeling

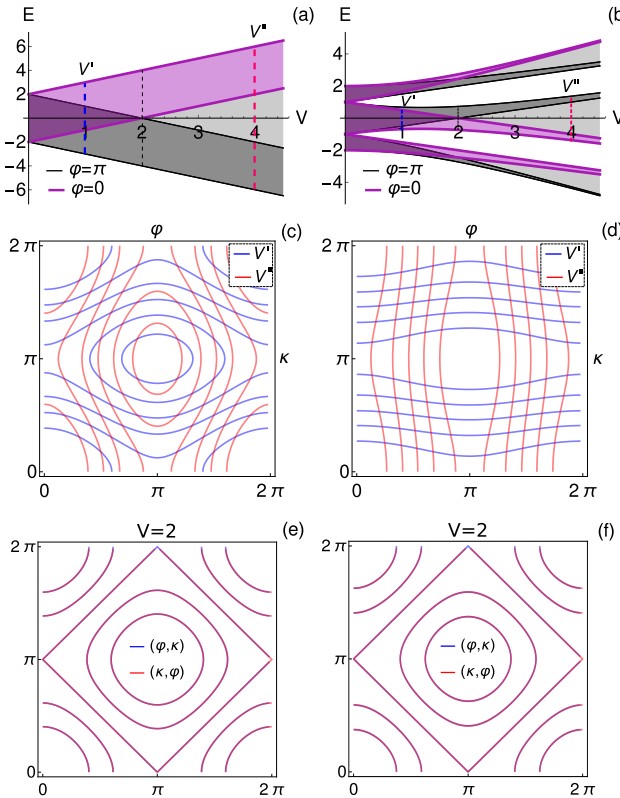

Figure 1: (a,b) Energy bands for commensurate approximants of the AAM, Eq. 1. Results for $\varphi = 0$ (purple shaded) and $\varphi = \pi$ (dark grey shaded) and integrated over $\kappa$ for $\tau_c = 1$ (a) and $\tau_c = 2/3$ (b), corresponding respectively to one and three sites per unit cell. The light grey shading at $V > 2$ denotes the energies that are swept when $\varphi$ is varied between $\varphi = 0$ and $\varphi = \pi$. (c,d) Generalized Fermi surfaces in the $(\varphi, \kappa)$ plane for the dual parameters $V' = 1$ and $V'' = 4$, for the energy bands indicated in (a,b) respectively. The results in Figs. (c),(d) are for $\tau_c = 1$ and $\tau_c = 2/3$, respectively. Note that the contours obtained for $V = V'$ are the same as the ones obtained for $V = V''$ if $\varphi$ and $\kappa$ are interchanged. (e,f) FS obtained at the self-dual point $V = 2$ for $\varphi$ ($\kappa$) in the horizontal axis, in blue (red). The results in Figs. (e),(f) are for $\tau_c = 1$ and $\tau_c = 2/3$, respectively. Note that the contours are self-dual under switching $\varphi$ and $\kappa$.

of the sites). As we did for the AAM, we encode this slide periodicity into a rescaled shift $\varphi \in ]-\pi, \pi]$, such that the system is invariant under $\varphi \to \varphi + 2\pi$.

We found that hidden dualities between the variables $\kappa$ and $\varphi$ of CA's are at the root of localization-delocalization transitions in 1D QPS. The identification of such dualities allows to determine mobility edges. As the size of the CA unit cell increases the mobility edges approach the ones of the true QPS. Surprisingly, CA with relatively small unit cells already provide an excellent approximation to the mobility edges of the limiting QPS.

As for the AAM, the hidden dualities can be identified through the generalized energy bands $E(\varphi, \kappa)$. Generalized duality transformations can be obtained for each band $E(\varphi, \kappa)$, as depicted in Fig. 2(e). There, we see that the $\kappa$-dependent energy dispersion in the extended phase of the limiting QPS is dual of the $\varphi$-dependent energy dispersion in the localized phase. This becomes even more clear in Fig. 2(f), where we show the generalized FS. This feature is a consequence of the duality transformation that maps wave functions at points in the extended phase to wave functions at their dual points in the localized phase, exchanging the

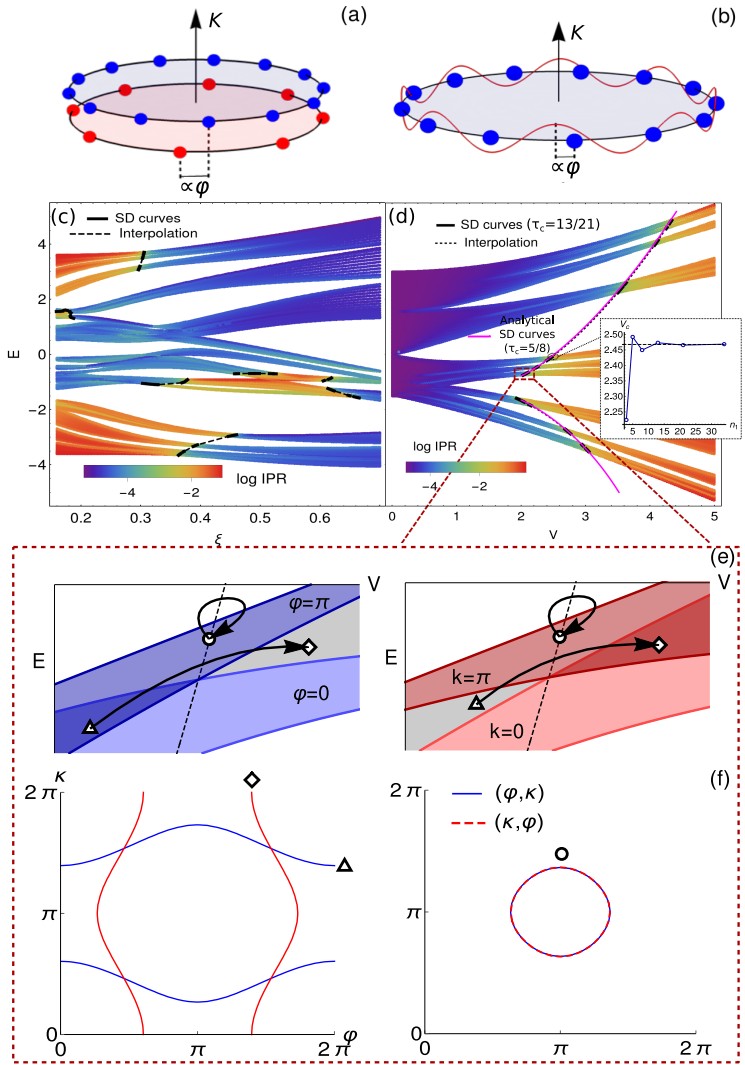

Figure 2: In this figure we approach a limiting QPS with $\tau = (\sqrt{5}-1)/2$. (a,b) Illustration of the dual parameters $k$ and $\phi$ for a periodic ladder (a) and a periodic chain (b). These examples are rings containing unit cells of CA with $\tau_c = 8/13$. (c,d) Numerical results for the energy spectrum and IPR of an EAAM (LM with gaussian decaying interlayer hoppings) as a function of potential's strength $V$ (interlayer hopping's decay length $\xi$), for $\tau_i = 89/144$. EAAM: Aubry-André model with additional next-nearest-neighbor hopping $t_2 = 0.5$, Eq. 9; LM: Ladder model with interlayer hopping strength $V = 3$ and cutoff $\Lambda = 2.5a$, Eq. 10. In black, we plot curves of SD points for the CA with $\tau_c = 13/21$ (obtained numerically) and in magenta, approximate analytical SD curves using the CA with $\tau_c = 5/8$. The inset contains the comparison between the SD points computed for CA with $\tau_c = F_n/F_{n+1}$ and with unit cell of size $n_1 = F_{n+1}$, with $F_n$ the Fibonnacci number of order $n$, and the estimation of the critical point through the IPR results for a system with $\tau = 4181/6765$ and $L = 6765$ sites. The definitions for the parameters here mentioned, including the IPR, $\tau_i$ and model parameters, are given in Sec. 4. (e) Close-up of an energy band for $\tau_c = 13/21$ [the range of parameters shown is contained in the red dashed box in (d)]. The energy bands obtained for $\varphi = 0, \pi$ ($k = 0, \pi$) and integrated over $\kappa$ ($\varphi$) are shown in blue (red) to illustrate the duality encoded in $\kappa, \varphi$. A generalized duality transformation for a CA maps points in the extended and localized phases of the limiting QPS (dual points, e.g. triangle and diamond markers), switching the roles of phases $\kappa$ and $\varphi$. The fixed points of this transformation (SD points, e.g. circle marker) are the manifestation of the mobility edge of the limiting QPS. (f) FS in the $(\varphi, \kappa)$ plane for the points indicated in (e). For the SD point (circle) the FS is shown with the standard, $(\varphi, \kappa)$, and switched, $(\kappa, \varphi)$, axis in order to show their perfect agreement.

roles of phases $\varphi$ and $\kappa$, just like we have seen for the AAM. The generalized FS of $E(\varphi, \kappa)$ at these dual points are related by a suitable interchange of $\varphi$ and $\kappa$. At the fixed points of the duality transformation, the FS are invariant under this interchange. Such self-dual (SD) points are the manifestation of the critical points of the limiting QPS in its CA. In the limit that $\tau_c \rightarrow \tau$ (infinite-size unit cell), the duality transformation of the CA reduces to the duality transformation of the limiting QPS.

The lines of SD points for CA provide remarkably accurate descriptions of the mobility edges of limiting QPS, in many cases even for CA with relatively small unit cells. Examples are shown in Fig. 2(c,d) for an EAAM and a LM. The SD curves obtained for a CA with $\tau_c = 13/21$ match very accurately the mobility edge separating extended (blue) and localized (red) phases obtained numerically via IPR. In the inset of Fig. 2(d) we can also see that the predictions of SD points converge to the critical point computed through the IPR using large system sizes for CA with relatively small unit cell size. Moreover, it is possible in some regimes to compute very accurate approximate analytical SD curves for CA with smaller unit cells, as shown in Fig. 2(d) for $\tau_c = 5/8$.

For the models with exact mobility edges in Refs. [34, 52], the SD curves are exact for any CA and CA-independent, perfectly matching the mobility edges of the limiting QPS. In these cases, the mobility edge can be obtained through the simplest possible CA. Additionally, we show in Sec. 6 that it is possible to use CA to define a generalized duality transformation that maps eigenstates of QPS at dual or SD points.

Our findings establish a strong connection between the localization-delocalization transitions in widely different 1D QPS in terms of dualities that are believed to be absent away from fine-tuned models like the AAM. They not only provide a simple way to characterize the phase diagrams of these systems in terms of CA, but also to understand whether a simple duality transformation can be defined for a given model.

# 3 Aubry-André Model for Commensurate Structures

In this section we review and prove the existence of the Aubry-André duality for commensurate structures, to justify our claims in the previous section. We again take the commensurate Aubry-André Hamiltonian in Eq. 1 by setting $\tau = \tau_c = n_2/n_1$, but now consider a periodic system formed by $N$ supercells. Expressing the label of each site as $n = m + r n_1$, where $m = 0, ..., n_1 - 1$ runs over sites within a unit cell and $r = 0, ..., N - 1$ runs over supercells, we can write the electron destruction operator as

$$c_n \equiv c_{r,m} = \frac{1}{\sqrt{N}} \sum_k e^{ik(n_1 r + m)} \tilde{c}_m(k), \qquad (2)$$

where $k = 2\pi j/(n_1 N)$, with $j = 0, ..., N - 1$, is the Bloch momentum. With this change, the Hamiltonian becomes block diagonal, $H = \sum_k H(\phi, k)$, where $H(\phi, k)$ depends parametrically on $k$ and $\phi$:

$$\tilde{H}(\phi, k) = t \sum_{m=0}^{n_1 - 1} \left( e^{-ik} \tilde{c}_{m+1}^\dagger(k) \tilde{c}_m(k) + e^{ik} \tilde{c}_m^\dagger(k) \tilde{c}_{m+1}(k) \right) \qquad (3)$$

$$+ \sum_{m=0}^{n_1 - 1} V \cos(2\pi \tau_c m + \phi) \tilde{c}_m^\dagger(k) \tilde{c}_m(k), \qquad (4)$$

where we used the fact that for $\tau_c = n_2/n_1$, $\cos(2\pi \tau_c (m + r n_1) + \phi) = \cos(2\pi \tau_c m + \phi)$. It can be seen that $H(\phi, k)$ is left invariant (up to a relabeling of the sites) under a change

$\phi \rightarrow \phi + 2\pi/n_1$, and (up to a gauge transformation) under a change $k \rightarrow k + 2\pi/n_1$. This allows us to defined the rescaled momenta and shift as $\kappa = n_1 k$ and $\varphi = n_1 \phi$, respectively. Writing the eigenstates of $H(\phi, k)$ as

$$|\psi\rangle = \sum_m \psi_m^{\rm r}(\phi,k)\tilde{c}_m^{\dagger}(k)|0\rangle, \tag{5}$$

the amplitudes $\psi_m^{\rm r}(\phi,k)$, where r stands for real space, satisfy the equation

$$te^{-ik}\psi_{m-1}^{\rm r}(\phi,k) + te^{ik}\psi_{m+1}^{\rm r}(\phi,k) + V\cos(2\pi\tau_c m + \phi)\psi_m^{\rm r}(\phi,k) = E\psi_m^{\rm r}(\phi,k), \tag{6}$$

where the energies depend parametrically on $\varphi$ and $\kappa$, $E = E(\varphi, \kappa)$. It is easy to see that the hopping term would be diagonalized with a further Fourier expansion in the $m$ indices. However, by writing the Aubry-André potential as $V\cos(2\pi\tau_c m + \phi) = V\cos(gx_m + \phi)$, where $x_m = m$ is the position of site $m$ (in units of $a_1$) and $g = 2\pi\tau_c$ (in units of $a_1^{-1}$) is the wavenumber of the potential, we can see that it will couple different momentum states by increments of $\pm g$. This motivates the following transformation

$$\psi_m^{\rm r}(\phi,k) = \frac{1}{\sqrt{n_1}}\sum_{q=0}^{n_1-1} e^{i2\pi\tau_c qm}\psi_q^{\rm d}(\phi,k), \tag{7}$$

where d stands for dual. The equation for the amplitudes $\psi_q^{\rm d}(\phi,k)$ thus becomes

$$2t\cos(2\pi\tau_c q + k)\psi_q^{\rm d}(\phi,k) + \frac{V}{2}e^{i\phi}\psi_{q-1}^{\rm d}(\phi,k) + \frac{V}{2}e^{-i\phi}\psi_{q+1}^{\rm d}(\phi,k) = E\psi_q^{\rm d}(\phi,k). \tag{8}$$

Comparing Eqs. (6) with (8), we see that under the transformation Eq. (7), the roles played by $k$ and $\phi$ (or by $\kappa$ and $\varphi$) are exchanged. Furthermore, if $V = 2t$ the model becomes self-dual. It is this duality that is at the heart of the localization-delocalization transition in the Aubry-André model [4]. In this work, we will see how generalized hidden dualities based on the dual roles played by rescaled shift and momentum, $\varphi$ and $\kappa$, can be used to determine localization-delocalization transition in general one-dimensional QPS.

## 4 Models and Methods

For the remainder of this work, we will focus on EAAM and LM introduced previously. Explicitly, the Hamiltonian for the family of EAAM is

$$H = \sum_n \mathcal{V}(2\pi\tau n + \phi)c_n^{\dagger}c_n + \sum_n \left(c_n^{\dagger}c_{n+1} + t_2 c_n^{\dagger}c_{n+2} + {\rm h.c.}\right), \tag{9}$$

where the first term contains the quasiperiodic potential, with $\mathcal{V}(x)$ a $2\pi$-periodic function, and the second term contains nearest-neighbor and next-nearest-neighbor hoppings. The ratio between the lattice constant and the wave length of the quasiperiodic potential is given by $\tau$, and the phase $\phi$ fixes the shift with respect to the first lattice site [see Fig. 3(a)]. For the standard AAM, $\mathcal{V}(x) = V\cos(x)$ and $t_2 = 0$. We also consider a modified AAM (MAAM) [34], where $\mathcal{V}(x) = 2V\cos(x)/[1 - \alpha\cos(x)]$ and $t_2 = 0$. In both models, $V$ is the strength of the quasiperiodic potential. The latter model has an exact mobility edge, in contrast with the AAM, for which the localization-delocalization transition occurs simultaneously for all energies at $|V| = 2$. Both models are self-dual under different duality transformations [4, 34], being associated with $\tau$-independent localization-delocalization transitions.

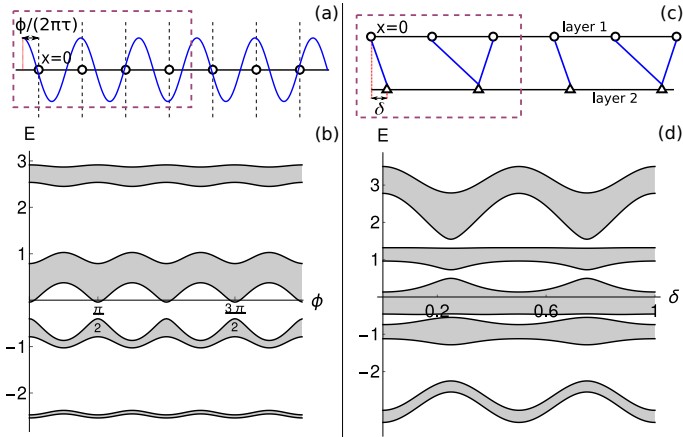

Figure 3: (a) Sketch of a commensurate EAAM, Eq. 9. In blue we represent the potential $\mathcal{V}(2\pi\tau n + \phi)$ for $\tau_c = n_2/n_1 = 3/4$. In this case, the unit cell depicted by the purple dashed box is composed by 4 sites. (b) Band edges for the commensurate system in (a) as a function of phase $\phi$. We considered the AA-NNN, with next-nearest-neighbor hoppings of strength $t_2 = 0.2$ and potential of strength $V = 1.75$. (c) Sketch of the LM, Eq. 10, for $\tau_c = n_2/n_1 = 2/3$. In this case, the unit cell, depicted by the purple dashed box, has 5 sites. (d) Band edges for the commensurate system in (c) as a function of the displacement $\delta$ defined in there. The parameters used for the LM were $V = 2.5$ (interlayer hopping's strength) and $\xi = 0.25$ (interlayer hopping's decay length).

If we consider $t_2 \neq 0$, the special self-duality of the previous models is broken [31]. In this case, the exact mobility edge is not known. Therefore, we also consider a next-nearest-neighbor AA model (AA-NNN), with $\mathcal{V}(x) = V\cos(x)$ and $t_2 \neq 0$. The EAAM results shown in Fig. 2(d) and 2(f) are for the AA-NNN with $t_2 = 0.5$.

In the LM, two chains with commensurate or incommensurate lattice constants are coupled by hopping terms [see Fig. 3(c)]. The Hamiltonian can be written as

$$H = \sum_{l,n} c_{ln}^{\dagger} c_{ln} + \sum_{|x_{1n} - x_{2m}| < \Lambda} t_{\perp}(|x_{1n} - x_{2m}|) c_{1n}^{\dagger} c_{2m} + \text{h.c.}, \tag{10}$$

where $c_{ln}^{\dagger}$ creates an electron at site $n$ of chain $l$ and $x_{1n} = a_1 n$ and $x_2 = a_2 n + a_1 \delta$ is the position of this site in an axis parallel to the layer, with $\delta$ a shift of layer 2 with respect to layer 1 [see Fig. 3(b)]. We consider $t_{\perp}(x) = V\exp(-x^2/\xi^2)$, where $V$ and $\xi$ are respectively the interlayer hopping strength and decay length, and again $\tau = a_1/a_2$. This model is associated with non-trivial mobility edges for some choices of the parameters. In what follows, we set $\Lambda = 2.5 a_1$ and $a_1 = 1$.

In order to capture the phase diagram of 1D QPS we analyse its CA, characterized by the rational $\tau_c = n_2/n_1$. The set of possible $\tau_c$ are the so-called convergents of the irrational number $\tau$, that can be computed through its continued fraction expansion. Convergents with larger $n_1$ approximate $\tau$ more accurately. Therefore, we label such convergents as *higher order approximants*. The unit cell of CA contains $n_1$ and $n_1 + n_2$ sites, respectively for the EAAM and the LM, as exemplified in Figs. 3(a,c).

As discussed in Sec. 3, for a given CA, changing $\phi \rightarrow \phi + 2\pi/n_1$ in Eq. (9) corresponds only to a redefinition of the unit cell. This follows from the coprimality between $n_1$ and $n_2$. Therefore, the Hamiltonian is $\phi$-periodic with period $2\pi/n_1$ [see Fig. 3(b)]. With this in mind, we define the rescaled shift as $\varphi = n_1\phi$. A single period of the energy dispersion is covered for $\varphi \in [0, 2\pi[$. In a similar way, we can note that for the LM, the unit cell is redefined under

$\delta \rightarrow \delta + 1/n_2$ [see Fig. 3(d)]. In this case, we define the rescaled shift as $\varphi = 2\pi\tau_c\delta$. Note that for more generic models, the definition of $\varphi$ can change - it should be always bounded by the $\phi(\delta)$-dependent periodicity of the energy bands (see Sec. 7 and Appendix C). Notice that the phase $\phi$ in an EAAM can also be interpreted as a shift $\delta$, by writing the potential as $\mathcal{V}(2\pi\tau_c n + \phi) = \mathcal{V}(2\pi\tau_c(n - \delta))$, where $\delta = -\phi/(2\pi\tau_c)$.

In order to characterize the localization properties of eigenstates of QPS, we also compute the inverse participation ratio (IPR) for large systems. This quantity is defined for each eigenstate $|\psi(E)\rangle = \sum_n \psi_n(E)|n\rangle$, where $\{|n\rangle\}$ is a basis localized at each site, as

$$\text{IPR}(E) = \frac{\sum_n |\psi_n(E)|^4}{(\sum_n |\psi_n(E)|^2)^2}. \tag{11}$$

For extended states, we expect $\text{IPR}(E) \sim L^{-1}$, where $L$ is the number of sites in the system. For localized states, $\text{IPR}(E) \sim$ constant. For a large enough system, the value of the IPR in the localized phase is significantly larger than in the extended phase. In practice, in order to simulate numerically the IPR of incommensurate systems, one must consider finite lattices with periodic boundary conditions in order to avoid defects. Thus, we approximate the incommensurate system by simulating a single unit cell with periodic boundary conditions of a CA with $\tau_i = n_2^i/n_1^i$, for very large $n_1^i$. Hereinafter, and even though we will always be considering CA's, we will denote the commensurability parameter for *high order approximants* as $\tau_i$, which will be considered when studying IPR; and reserve $\tau_c$ for the parameter of a CA with moderate size, which will be used to construct generalized hidden dualities.

## 5 Spectral duality

In this section, we study the energy bands of CA as a function of wave vector $\kappa$ and phase $\varphi$. The aim is ultimately to obtain the phase diagram of the limiting QPS, including possible mobility edges.

**Far from critical point.—** We start by plotting in Fig. 4 the band edges of some CA on top of the phase diagram obtained for the AA-NNN [shown in Fig. 2(d)], for a selected region of parameters around the critical point. The band edges are obtained for the energy bands $E_n(\varphi, k)$ that appear in this region of parameters, for fixed $\varphi = 0, \pi$. In the shown examples, the energy bands for all $\varphi \in [0, 2\pi[$ are bounded by the lowest and highest energy band edges of $\varphi = 0$ and $\varphi = \pi$. In Fig. 4(a), we can see that for a CA with $\tau_c = 21/34$, these band edges bound very well the energy intervals over which a finite density of states (DOS) is observed for the QPS. However, if we zoom in to a narrower region of parameters around the critical point, we see in Fig. 4(b)-left that the bounds are not accurate for $\tau_c = 21/34$ (some band edges do not bound any spectral weight, as exemplified with vertical double arrows). A higher-order approximant is needed for an accurate bounding, as seen in Fig. 4(b)-right for $\tau_c = 55/89$. In fact, the closer we are to the critical point, the larger the CA's order needed for an accurate bounding.

We illustrate how the energy bands of CA behave in the extended and localized phases of the limiting QPS using the AA-NNN. Fig. 5 depict the energy bands $E_n(\varphi, \kappa)$ for CA of the QPS defined by $\tau = 0.418(\sqrt{5} + 1)/2 = 0.676338\cdots$. Figs. 5(a,c), show the band edges when $\varphi = 0$ (blue lines) and $\varphi = \pi$ (black lines) for two CA systems. Filled regions in blue ($\varphi = 0$) or dark grey ($\varphi = \pi$) correspond to states inside the respective band, while light grey regions indicate all the states that would appear for $0 < \varphi < \pi$. The following observations can be made:

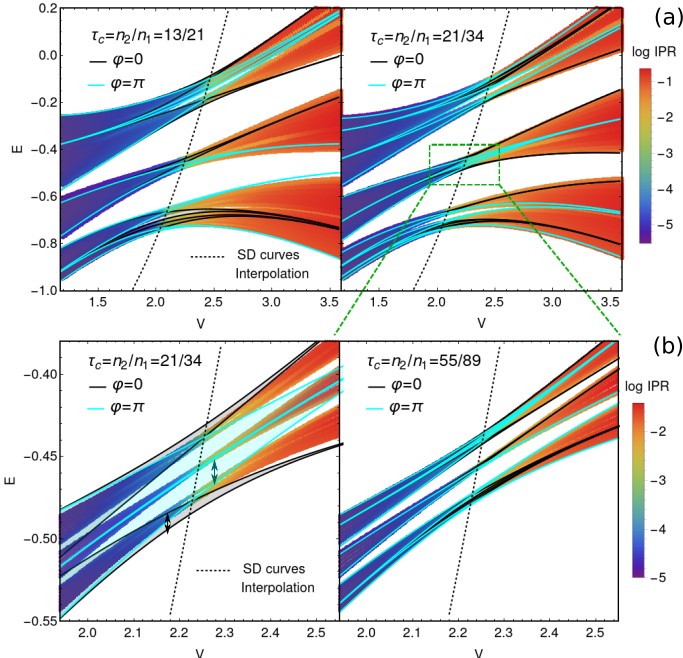

Figure 4: (a) Band edges for $\varphi = 0$ and $\varphi = \pi$, for CA with $\tau_c = 13/21$ (left) and $\tau_c = 21/34$ (right), along with the IPR for $\tau_i = 233/377$. The light gray and light cyan shading denotes the range of energies delimited by the band edges of $\varphi = 0$ and $\varphi = \pi$ respectively. Results are for the AA-NNN defined in Eq. 9 and below it, with next-nearest-neighbor hopping $t_2 = 0.5$ (close-up of Fig. 2(d) for a selected region of parameters around the critical point). (b) Same as (a), but for a narrower region of parameters around criticality encompassed by the green dashed box in (a)-right, and for $\tau_c = 21/34$ (left) and $\tau_c = 55/89$ (right). The arrows exemplify band edges that poorly approximate the region of finite spectral weight.

1. The energy bands of CA weakly depend on $\varphi$ ($\kappa$) in the extended (localized) phases of the limiting QPS. On the other hand, they strongly depend on $\kappa$ ($\varphi$) in the extended (localized) phases.

2. The energy bands depend equally on $\varphi$ and $\kappa$ at the SD points;

3. For higher-order approximants, the $\varphi$-dependence ($\kappa$-dependence) of the energy dispersion decreases abruptly in the extended (localized) phase. In fact, this decay is exponential in the size of the unit cell, with a characteristic length scale that correspond to a possible definition of the correlation length in extended (localized) phase (see Appendix A for details).

Observations 1-to-3 can also be made looking at constant-energy cuts. Examples are shown in Figs. 5(b,d), where the FS is observed in points for which $E_n(\varphi, \kappa)$ depends weakly either on $\varphi$ or $\kappa$. The FS corresponds to almost straight lines for constant $\varphi$ ($\kappa$) and is weakly dependent on $\kappa$ ($\varphi$).

The region of parameters for which $E_n(\varphi, \kappa)$ depends significantly both on $\kappa$ and $\varphi$ shrinks around the SD points as we increase the order of the CA. In the limit of high-order approximants, there is essentially no $\varphi$-dependence ($\kappa$-dependence) in the extended (localized) phase except for this narrowing region, that ultimately reduces to the SD point in the limit $\tau_c \to \tau$.

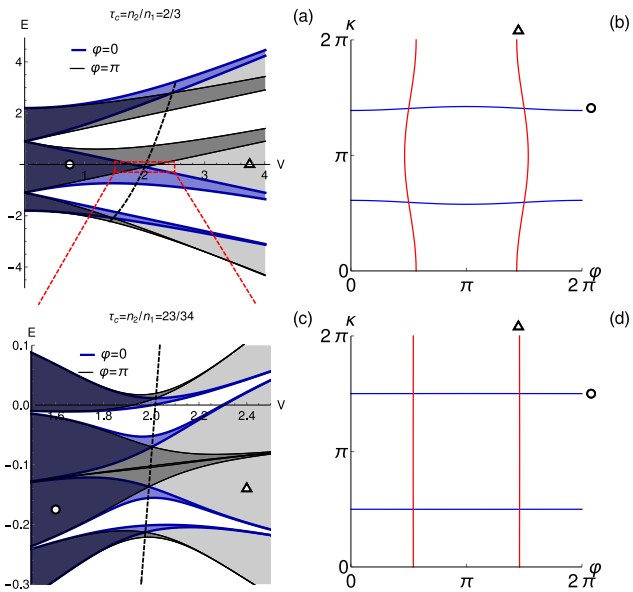

Figure 5: Energy bands for CA of the irrational $\tau = 0.418(\sqrt{5}+1)/2 = 0.676338\cdots$, for the AA-NNN defined in Eq. 9 and below it, with next-nearest-neighbor hopping $t_2 = 0.1$. (a,c) Band edges for the commensurate approximants $\tau_c = n_2/n_1 = 2/3$ (a) and $\tau_c = n_2/n_1 = 23/34$ (c), for $\varphi = 0, \pi$. In addition, we plot the SD curves in dashed black (we interpolate these curves inbetween bands). Note that the range of energies and $V$ depicted in (c) is contained in the dashed red box shown in (a). (b,d) FS in the $(\varphi, \kappa)$ plane of generalized energy bands $E(\varphi, \kappa)$, for the points marked in (a,c).

**Close to critical point.—** We now explore the regions in parameter space near the critical SD points.

Fig. 6(a) show the band edges for the CA defined by $\tau_c = 13/21$, for $\varphi = 0, \pi$. Together with the band edges, we plot a set of SD curves in Fig. 6(a). Along these curves, the FS is invariant under a suitable interchange of $\varphi$ and $\kappa$, to an almost perfect approximation. In fact, the perfect invariance only arises when $\tau_c \to \tau$ (see Appendices B.2 and C), but in practice it can be seen even for low-order CA. The transformation that interchanges $\varphi \leftrightarrow \kappa$, and that we denote $\mathcal{R}_0$, corresponds to a $\pi/2$-rotation in the $(\varphi, \kappa)$ plane around some point $(\varphi_0, \kappa_0)$. To better illustrate this point, we zoomed into a generalized band $E(\varphi, \kappa)$ as shown in the bottom panel of Fig. 6(a). Note that the $E(\varphi = 0, \kappa)$ and $E(\varphi = \pi, \kappa)$ bands split at a point contained in the SD curve. This feature is generic. Fig. 6(b) depicts the FS for different parameters specified by points in the bottom panel of Fig. 6(a). At the SD points (middle panel), the FS is invariant under $\mathcal{R}_0$. Figure 6(b) also suggests that FS for $E > E_c(V)$ can be mapped into FS for $E < E_c(V)$ upon this rotation, with $E_c(V)$ the energy at the SD point: the FS at points with $E < E_c(V)$ (blue curves in leftmost sub-figures of Fig. 6(b) ) are identical to the rotated FS at the corresponding dual points at $E > E_c(V)$ (red dashed curves in rightmost sub-figures of Fig. 6(b) ). Such observation hints at the existence of a generalized Aubry-André duality between the extended and localized phases that switches the roles of phases $\phi$ and $k$, as in the AAM [4].

Remarkably, the SD curves for a fixed CA, obtained by requiring invariance under $\mathcal{R}_0$, approximate unexpectedly well the mobility edge of the QPS, as seen in Fig. 2(d). This approximation becomes increasingly better as the order of the order of the approximant increases. Convergence can be controlled by comparing the results for two consecutive CA (see Fig. 24 in Appendix C for examples).

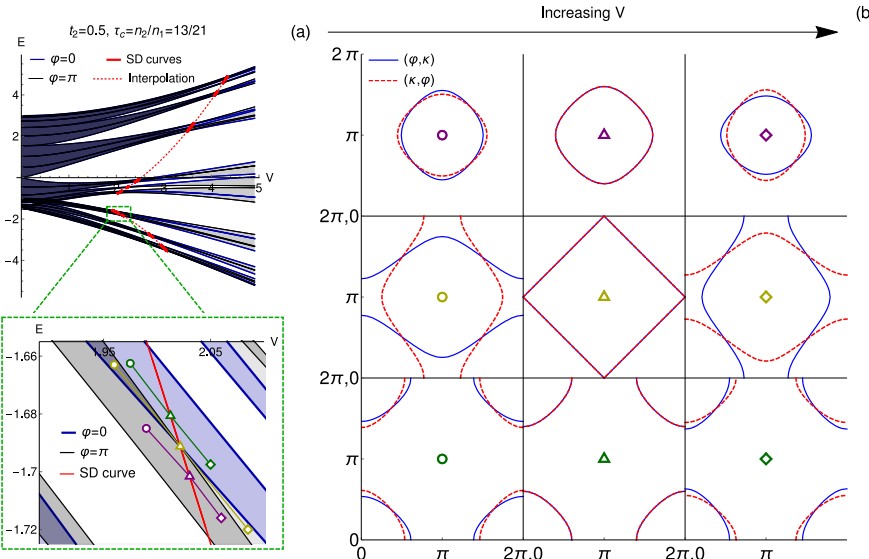

Figure 6: Example of spectral duality for the AA-NNN, defined in Eq. 9 and below it. (a) Top: Band edges for the commensurate approximant $\tau_c = n_2/n_1 = 13/21$, for $\varphi = 0, \pi$ and next-nearest-neighbor hoppings $t_2 = 0.5$, along with the SD curves. Bottom: Close-up of randomly chosen bands inside the green dashed box in the upper figure. The qualitative behaviour of the $\varphi = 0, \pi$ bands is always the same as depicted here: the bands split at a point within the SD curve. (b) FS for the points depicted at the bottom of Fig. (a). The point corresponding to each FS is shown at the center of each figure. We consider $\varphi$ ($\kappa$) in the $x$ ($y$)-axis for the full blue (dashed red) curves. In the SD points, the FS are invariant under a $\pi/2$-rotation $\mathcal{R}_0$ around $(\varphi_0, \kappa_0) = (0, 0)$ [or $(\varphi_0, \kappa_0) = (\pi, \pi)$], which corresponds to interchanging $\kappa \longleftrightarrow \varphi$.

For all the systems that we tested, the SD curves always approximate very well the mobility edge, even in regimes where a numerical analysis based on the IPR can fail [2]. A generic example for the LM can be seen in Fig. 2(c), for which the phase diagram is highly non-trivial. Remarkably, the FS close-enough to criticality are always of the form observed in Fig. 6(b), for CA of high enough order and irrespectively of the studied model. In Appendices B.2 and C we show examples of additional models and provide details on the computation of SD points.

The results we presented up to now naturally raise the question of whether a hidden duality transformation generalizing that found by AA exist for a generic localization-delocalization transitions in 1D QPS. We make the case for its existence in Sec. 6 . However, it is worth noting that in general such duality may not be well defined for low-order approximants. As an example, consider the FS for CA of the AA-NNN at fixed $V = 2.1$ and $t_2 = 0.2$, and variable $E$ [see Fig. 7]. For $\tau_c = 1$ there is no SD energy for which the FS is invariant under the $\varphi \longleftrightarrow \kappa$ interchange, as seen in the top row panel of Fig. 7. For the higher-order approximant shown in the bottom row panel, $\tau_c = 2/3$, the invariance seems to occur for $E_c \approx 0.07$. In fact, a more detailed analysis shows that the invariance is not yet perfect for this approximant, but becomes increasingly better for higher-order approximants (see Appendix B.2). Indeed, for all the models studied, we observe that the FS always acquires the simple shapes shown in Fig. 6(b), close to the critical point and for CA of high-enough order.

---

[2]When the IPR is computed for QPS with incommensurate ratio $\tau$ very close to commensurate ratios $\tau_c$ of low-order CA, it can give wrong results. In particular, we can have IPR $\sim$ constant in the extended phase. This problem occurs for very weakly dispersive states, in particular when the energy dispersion both in $\varphi$ and $\kappa$ is below machine precision. It can occur for any model, including the AAM.

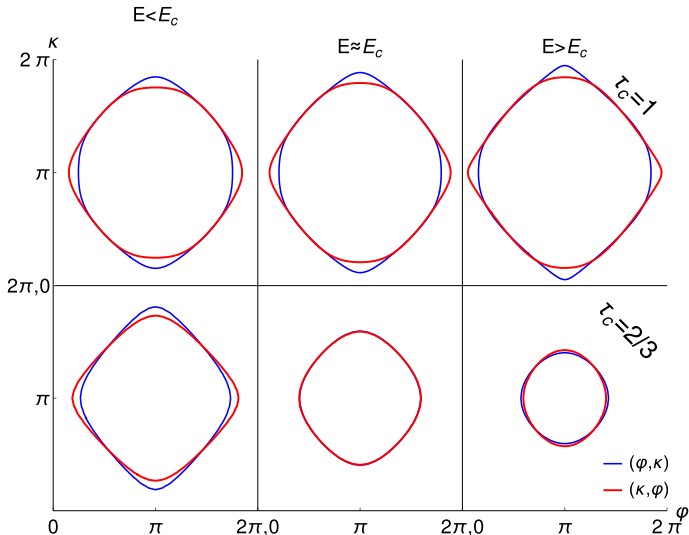

Figure 7: Constant energy cuts of the energy bands $E_n(\varphi, \kappa)$ for the AA-NNN, defined in Eq. 9 and below it, with fixed $V = 2.1$, $t_2 = 0.2$ and variable $E$. The results are for the CA defined by $\tau_c = 1$ (top row) and $\tau_c = 2/3$ (bottom row).

**Parametrization of constant energy curves near SD points—** For a higher enough order of the approximant, the functional form of the FS as a function of $\varphi$ and $\kappa$ can be well captured by sinusoidal shapes, close enough to quasiperiodicity-driven localization-delocalization transitions. Considering an Hamiltonian with parameters $\boldsymbol{\lambda}$, we have at fixed energy $E$ (see Appendix B for details):

$$E = V_R(\boldsymbol{\lambda}, E)\cos(\varphi - \varphi_0) + 2t_R(\boldsymbol{\lambda}, E)\cos(\kappa - \kappa_0) + E_R(\boldsymbol{\lambda}, E). \qquad (12)$$

This is a generalization of the renormalized model defined in Ref. [37] for the AAM, encoded in the energy-dependence of the parameters. $V_R$, $t_R$ and $E_R$ are renormalized couplings independent of $\varphi$ and $\kappa$.

The ansatz of Eq. (12) assumes that for large enough unit cells only the fundamental harmonics in $\varphi$ and $\kappa$ survive. In fact $V_R$ ($t_R$) also becomes irrelevant, i.e. vanishes, in the extended (localized) phase as the CA's order is increased. This assumption is motivated phenomenologically here by the universality observed in the FS of different models and in the next section by some special models where Eq. (12) is exact. A deeper insight on its validity will be provided elsewhere [54].

According to our hypothesis, in the extended (localized) phase the sub-bands of a CA of high-enough order can be described by a renormalized $\kappa$-dependent ($\varphi$-dependent) single-band effective model. In that case, further increasing the CA's order approximately folds the model's energy band. Examples of the "band-folding" in the extended and localized phases are given in Fig. 8.

At the critical point, both $V_R$ and $t_R$ are relevant: energy gaps are opened irrespectively of the CA's order.

This hypothesis also explains why the regions of finite DOS for a QPS are so well bounded by the energy bands of its CA in the extended and localized phases: the band edges are not significantly changed by the approximated band-foldings. Furthermore, the accuracy of the commensurate approximation increases quadratically with the unit cell's size. To understand why, consider a CA defined by $\tau_c^m = n_2^m / n_1^m$, as the $m$-th order convergent of the irrational $\tau$ [55]. A well-known property of the convergents is that $|\tau_c^m - \tau| < (n_1^m n_1^{m+1})^{-1}$ [55]. Therefore, for CA corresponding to $\tau_c^m$ and the QPS characterized by $\tau$ only have significant differences at

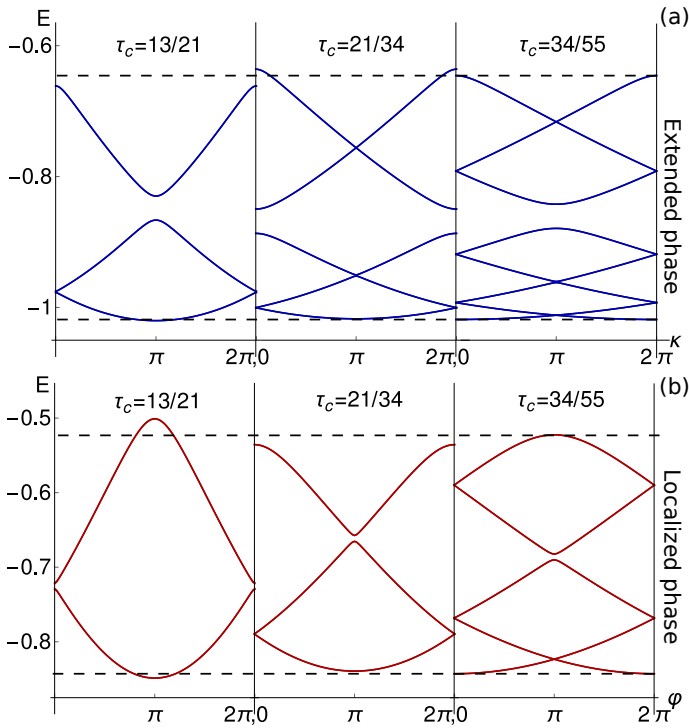

Figure 8: Examples of "band-folding" for some selected energy bands and different CA. These results are for the AA-NNN, defined in Eq. 9 and below it, with $t_2 = 0.5$. (a) Energy dispersion with $\kappa$ for $V = 1$ (extended phase) and three different CA. The dispersion with $\varphi$ is smaller than the line width. (b) Energy dispersion with $\varphi$ for $V = 3.5$ (localized phase) and three different CA. The dispersion with $\kappa$ is smaller than the line width. The dashed lines are the band edges of the lowest and highest energy bands shown, for $\tau_c = 34/55$. The energy-bands of higher-order CA are approximately band-foldings of the energy bands of lower-order ones.

length scales $L_m \sim n_1^m n_1^{m+1} > (n_1^m)^2$. On the other hand, in extended and localized phases, the correlation length, $\xi$, is finite. Therefore, for $L_m > \xi$, the CA provides a good approximation of the QPS: differences between both systems only arise at length scales larger than $\xi$. The fast convergence of the energy bands with the order of the approximate is a also consequence of the fast increase in $L_m > (n_1^m)^2$.

**Models with CA-independent self-dualities.—** Some special models exhibit a $\tau$-independent self-duality symmetry. These include for instance the AAM, the MAAM and the model in [52]. For these models, the $\varphi \leftrightarrow \kappa$ self-duality is exact independently of the CA. The localization-delocalization transition curve can then be easily computed analytically by analysing the simplest possible CA.

For the AAM, the simplest approximant has one site per unit cell ($\tau_c = 1$) and the single band energy dispersion is given by

$$E_{AAM}(V, \varphi, k) = V \cos\varphi + 2 \cos\kappa. \tag{13}$$

It is easy to see that for $V = 2$, the energy dispersion is invariant under $\varphi \leftrightarrow \kappa$. This interchanging operation is identical to a $\pi/2$-rotation, $\mathcal{R}_0$, around $(\varphi_0, \kappa_0) = (0, 0)$. We can also check for a rotation symmetry around $(\varphi_0, \kappa_0) = (\pi, 0)$. To do so, we redefine $\varphi' = \varphi + \pi$, yielding $E_{AAM}(V, \varphi', \kappa) = -V \cos\varphi' + 2 \cos\kappa$. The model becomes now self-dual under the new

rotation when $V = -2$. Putting the two conditions together, we get the well known critical potential (and self-dual condition) $|V| = 2$.

For the MAAM, the energy dispersion for the CA with $\tau_c = 1$ is

$$E_{\mathrm{MAAM}}(V, \alpha, \varphi, \kappa) = \frac{2V \cos \varphi}{1 - \alpha \cos \varphi} + 2 \cos \kappa. \tag{14}$$

Rearranging, we get $E = (\alpha E + 2V) \cos \varphi + 2 \cos \kappa - 2\alpha \cos \varphi \cos \kappa$, for $1 - \alpha \cos \varphi \neq 0$, which is self-dual under $\varphi \leftrightarrow \kappa$ if $E = 2(1-V)/\alpha$. Defining $\varphi' = \varphi + \pi$, as for the AAM, we get $E = -2(1+V)/\alpha$. Putting the two together, we get the condition for the mobility edge found in Ref. [34].

Finally, consider some of the models in Ref. [52], defined by

$$H = \sum_j (c_j^\dagger c_{j+1} + \mathrm{H.c.}) + 2 \sum_j \lambda_j c_j^\dagger c_j, \tag{15}$$

$$\lambda_j = \begin{cases} \lambda \cos(2\pi\tau j + \phi), & \mathrm{mod}\,(j, \nu) = 0, \\ 0, & \text{otherwise}, \end{cases} \tag{16}$$

where $\nu$ is an integer. For $\nu = 2$, the simplest possible approximant has two sites per unit cell, and the following characteristic polynomial:

$$\mathcal{P}(\kappa, \varphi) = C_{\mathrm{inv}}(\kappa, \varphi) - 2 \cos(\kappa) - 2E\lambda \cos(\varphi), \tag{17}$$

where $C_{\mathrm{inv}}(\kappa, \varphi) = E^2 - 2$ contains the terms that are invariant under switching $\kappa$ and $\varphi$. The phase invariance condition is therefore $E = \pm 1/\lambda$, which is the mobility edge expression in Eq. (5) of Ref. [52]. For $\nu = 3$, the simplest possible CA has three sites per unit cell and a characteristic polynomial:

$$\mathcal{P}(\kappa, \varphi) = C_{\mathrm{inv}}(\kappa, \varphi) + 2 \cos(\kappa) + 2(E^2 - 1)\lambda \cos(\varphi), \tag{18}$$

where $C_{\mathrm{inv}}(\kappa, \varphi) = 3E - E^3$, which gives the following condition for the mobility edge, $E = \pm\sqrt{1 \pm 1/\lambda}$, the same obtained in Ref. [52].

## 6 Generalized local duality transformation

Having established a generalized duality transformation in the spectrum, we now extend it to the eigenstates.

### 6.1 Eigenstates and symmetries in $(\phi, k)$ space

In this section we present the symmetries of the eigenstates of CA in the $(\varphi, \kappa)$ space. Our goal is to obtain the transformation properties of the real-space wave-function and its dual under translations in $k$ and $\phi$. This will then be used as a starting point for the definition of the generalized duality transformation in the follow-up section. To simplify the discussion, we restrict ourselves to EAAM in the following and later show that our description can also be applied to LM through an example. The Hamiltonian for this class of models can be written as

$$H = \sum_n \mathcal{V}(2\pi\tau_c n + \phi_0 + \phi) c_n^\dagger c_n + \sum_{n,n'=0} t(|n-n'|) e^{-ik_0(n-n')} c_n^\dagger c_{n'}, \tag{19}$$

where $\mathcal{V}(x)$ is a $2\pi$-periodic function, $t(|x|)$ is a generic function with $t(0) = 0$ and we introduced a phase twist $k_0$ in the hopping terms to further enlarge the class of models described by this expression. After applying the transformation in Eq. (2), this can be written as

$$
\tilde{H}(\phi, k) = \sum_{m=0}^{n_1-1} \mathcal{V}(2\pi\tau_c m + \phi_0 + \phi)\tilde{c}_m^\dagger(k)\tilde{c}_m(k) \\
+ \sum_{r\in\mathbb{Z}} \sum_{m,m'=0}^{n_1-1} t(|rn_1 + m - m'|)e^{-i(k+k_0)(m+rn_1-m')}\tilde{c}_m^\dagger(k)\tilde{c}_{m'}(k) .
\tag{20}
$$

The terms with $|r| \geq 1$ correspond to hopping terms between different unit cells.

**Properties of the real-space wave functions.** The eigenstates of $\tilde{H}(\phi, k)$ can be expanded as

$$
|\psi(k)\rangle = \sum_m \tilde{\psi}_m^r(\phi, k)\tilde{c}_m^\dagger(k)|0\rangle .
\tag{21}
$$

For the real-space wave-function $\tilde{\psi}_m^r(\phi, k)$, shifts in $\phi$ of $\phi_j = 2\pi j/n_1$ correspond to cyclical translations in $\tilde{\psi}_m^r$, that is,

$$
\tilde{\psi}_m^r(\phi + 2\pi j/n_1, k) = \tilde{\psi}^r_{\text{mod}(m+\Delta m_j, n_1)}(\phi, k).
\tag{22}
$$

This can be seen by absorbing $\phi_j$ in index $m$ of the potential $\mathcal{V}(2\pi\tau_c m + \phi_0 + \phi)$. In particular, we have that $2\pi\tau_c m + \phi_j = 2\pi\tau_c(m + \Delta m_j)$, where $\Delta m_j \equiv (j + ln_1)/n_2$ and $l$ is an integer such that $\Delta m_j$ is also an integer (the term with $l$ contributes with an irrelevant phase $2\pi l$ in the $2\pi$-periodic potential $\mathcal{V}(x)$). Furthermore, since $l$ satisfies the linear Diophantine equation $n_2\Delta m_j = j + ln_1$, it always has a solution since $n_1$ and $n_2$ are co-prime integers [56], see also [57]. Finally, since $\mathcal{V}(2\pi\tau_c(m + n_1) + \phi_0 + \phi) = \mathcal{V}(2\pi\tau_c m + \phi_0 + \phi)$, a translation in the potential of $m \to m + \Delta m_j$ corresponds to a cyclical translation of $\tilde{\psi}_m^r$ as written in Eq. 22. Therefore, as previously mentioned, shifts of $\phi_j$ correspond to redefinitions of the unit cell. There are $n_1$ possible such redefinitions that can be obtained through the different shifts $\phi_j$, with $j = 0, \cdots, n_1 - 1$.

On the other hand, under shifts in $k$ one obtains,

$$
\tilde{\psi}_m^r(\phi, k + k_j) = e^{-imk_j}\tilde{\psi}_m^r(\phi, k), \ k_j = 2\pi j/n_1 .
\tag{23}
$$

This can be seen by applying the gauge transformation $\tilde{c}_m^\dagger(k) = e^{im(k+k_0)}c_m^\dagger(k)$, such that

$$
\tilde{H}(\phi, k) = \sum_{m=0}^{n_1-1} \mathcal{V}(2\pi\tau_c m + \phi_0 + \phi)c_m^\dagger(k)c_m(k) \\
+ \sum_{r\in\mathbb{Z}} \sum_{m,m'=0}^{n_1-1} t(|rn_1 + m - m'|)e^{-irn_1(k+k_0)}c_m^\dagger(k)c_{m'}(k) .
\tag{24}
$$

In the new basis, $\tilde{H}$ is explicitly periodic under the transformations $k \to k + 2\pi j/n_1$, with $j \in \mathbb{Z}$ and setting $c_m^\dagger(k + 2\pi j/n_1) \to c_m^\dagger(k)$. Therefore, the new amplitudes $\psi_m^r(\phi, k)$ obeying

$$
|\psi(k)\rangle = \sum_m \psi_m^r(\phi, k)c_m^\dagger(k)|0\rangle ,
\tag{25}
$$

are also periodic $\psi_m^r(\phi, k + 2\pi j/n_1) = \psi_m^r(\phi, k)$. Using this equality and Eqs. 21,25 we can arrive at Eq. (23).

**Properties of the dual wave functions.** The dual wave function, $\tilde{\psi}_q^{\mathrm{d}}(\phi, k)$, can be defined as in the AA case in Eq. (32). The transformations of $\tilde{\psi}_q^{\mathrm{d}}(\phi, k)$ under shifts in $\phi$ of $k$ can also be obtained. Inverting relation Eq. (32) and using the Eq. (23) we obtain

$$\tilde{\psi}_q^{\mathrm{d}}(\phi, k + k_j) = \frac{1}{\sqrt{n_1}} \sum_{m=0}^{n_1-1} e^{-i2\pi\tau_c qm} \tilde{\psi}_m^{\mathrm{r}}(\phi, k + k_j) \tag{26}$$

$$= \frac{1}{\sqrt{n_1}} \sum_{m=0}^{n_1-1} e^{-im(2\pi\tau_c q + k_j)} \tilde{\psi}_m^{\mathrm{r}}(\phi, k). \tag{27}$$

From the right-hand side of this expression we can see that the term $k_j$ might be absorbed into $q$, yielding

$$\tilde{\psi}_q^{\mathrm{d}}(\phi, k + k_j) = \frac{1}{\sqrt{n_1}} \sum_{m=0}^{n_1-1} e^{-im2\pi\tau_c \left(q + \frac{j}{n_2} + l\frac{n_1}{n_2}\right)} \tilde{\psi}_m^{\mathrm{r}}(\phi, k) \tag{28}$$

$$= \tilde{\psi}^{\mathrm{d}}_{\mathrm{mod}\,(q+\Delta q_j, n_1)}(\phi, k), \tag{29}$$

where we again used that $k_j = 2\pi j/n_1$ and $l$ is an integer such that $\Delta q_j \equiv (j + ln_1)/n_2$ is also an integer, exactly as in the definition of $\Delta m_j$, below Eq. 22. Using the same arguments as for $\Delta m_j$, we can again always find $l$ that makes $\Delta q_j$ an integer. Finally, since $\tilde{\psi}_{q+n_1}^{\mathrm{d}} = \tilde{\psi}_q^{\mathrm{d}}$, we arrive at Eq. 29.

For $k_j = 2\pi j/n_1 \in [0, 2\pi[$, $\Delta q_j$ can take the values $\Delta q_j = 0, \cdots, n_1 - 1$. The different possible wave functions $\tilde{\psi}_q^{\mathrm{d}}(\phi, k + k_j)$ are therefore the $n_1$ possible cyclic translations of $\tilde{\psi}_q^{\mathrm{d}}(\phi, k)$.

**Cyclical translations under shifts in $k$ and $\phi$.** Defining the $n_1$-component vectors $\tilde{\boldsymbol{\psi}}^{\mathrm{r}}(\phi, k) = \left\{\tilde{\psi}_0^{\mathrm{r}}(\phi, k), \tilde{\psi}_1^{\mathrm{r}}(\phi, k), \dots\right\}^T$ and $\tilde{\boldsymbol{\psi}}^{\mathrm{d}}(\phi, k) = \left\{\tilde{\psi}_0^{\mathrm{d}}(\phi, k), \tilde{\psi}_1^{\mathrm{d}}(\phi, k), \dots\right\}^T$, we can summarize the previous results as

$$\tilde{\boldsymbol{\psi}}^{\mathrm{r}}(\phi + 2\pi j/n_1, k) = \mathcal{T}^{\Delta m_j} \tilde{\boldsymbol{\psi}}^{\mathrm{r}}(\phi, k), \tag{30}$$

$$\tilde{\boldsymbol{\psi}}^{\mathrm{d}}(\phi, k + 2\pi j/n_1) = \mathcal{T}^{\Delta q_j} \tilde{\boldsymbol{\psi}}^{\mathrm{d}}(\phi, k), \tag{31}$$

where $\mathcal{T}_{ij} = \delta_{i,\,\mathrm{mod}\,(j+1, n_1)}$ denotes the cyclical translation matrix, and $\Delta m_j$ and $\Delta q_j$ are the integers defined previously.

At this point, a duality between the wave functions in Eqs. 31,30, switching the roles of $\phi$ and $k$, is already apparent. In the following section, we will make use of this insight to define a duality mapping between these wave functions.

## 6.2 Definition of generalized duality transformation

**Motivation and definition.—** The results in section 5 for the energy bands of CA hinted at the existence of generalized dualities between the energy bands in the extended and localized phases. Here we complete our description by studying also the CA wave functions, and explicitly constructing a generalized duality transformation.

For a Hamiltonian depending on a set of parameters $\boldsymbol{\lambda}$, we can define dual points $P \equiv P\left(\boldsymbol{\lambda}, E(\boldsymbol{\lambda}, \varphi, \kappa)\right)$ and $P' \equiv P'\left(\boldsymbol{\lambda}', E'(\boldsymbol{\lambda}', \mathcal{R}_0[\varphi, \kappa]^T)\right)$ for each energy band of a CA such that the FS at $P$ is the rotation $\mathcal{R}_0$ of the FS at $P'$, see Fig. 9(a,b). SD points are defined by $P = P'$. Recall that $\mathcal{R}_0$ is a $\pi/2$-rotation in the $(\phi, k)$ plane, around point $(\phi_0, k_0)$.

These parameters were introduced in Eq. (20) and encode the fixed point of the duality transformation in the $(\phi, k)$ plane [which does not have to be $(\phi_0, k_0) = (0,0)$]. In the same way, as stated before, the FS of dual points in the phase diagram are identical under rotation $\mathcal{R}_0$ in the $(\varphi, \kappa)$ plane, around $(\varphi_0, \kappa_0)$. The latter are related with $(\phi_0, k_0)$ through $(\varphi_0, \kappa_0) = n_1\big(\ \mathrm{mod}\ (\phi_0, 2\pi/n_1),\ \ \mathrm{mod}\ (k_0, 2\pi/n_1)\big)$.

We can compare the real-space wave function vectors at point $P'$, $\tilde{\psi}^{\mathrm{r}}(P'; \mathcal{R}_0[\phi, k]^T)$, [3] with its Aubry-André dual at point $P$, $\tilde{\psi}^{\mathrm{d}}(P; \phi, k)$, with entries are given by

$$\tilde{\psi}_n^{\mathrm{d}}(P; \phi, k) = \frac{1}{\sqrt{n_1}} \sum_{m=0}^{n_1-1} e^{-i2\pi\tau_c mn} \tilde{\psi}_m^{\mathrm{r}}(P; \phi, k), \tag{32}$$

where, besides the dependence on $\phi$ and $k$, we also specified the points at which the wave functions are calculated. For the AAM, $\tilde{\psi}^{\mathrm{r}}(P'; \mathcal{R}_0[\phi, k]^T) = \tilde{\psi}^{\mathrm{d}}(P; \phi, k)$, as we have seen in Sec. 3, with $k_0 = 0$ and $\phi_0 = 0 \vee \phi_0 = \pi$. For generic models this is not the case. However, interestingly, they can be very similar. An example is shown in Fig. 9 (c) for the AA-NNN.

We can define the exact generalized duality transformation that maps all the wave functions at a point $P'$ to the Aubry-André dual wave functions at $P$, as simply the matrix transformation

$$\mathcal{O}_c \tilde{\psi}^{\mathrm{d}}(P; \phi, k) = \tilde{\psi}^{\mathrm{r}}(P'; \mathcal{R}_0[\phi, k]^T).$$

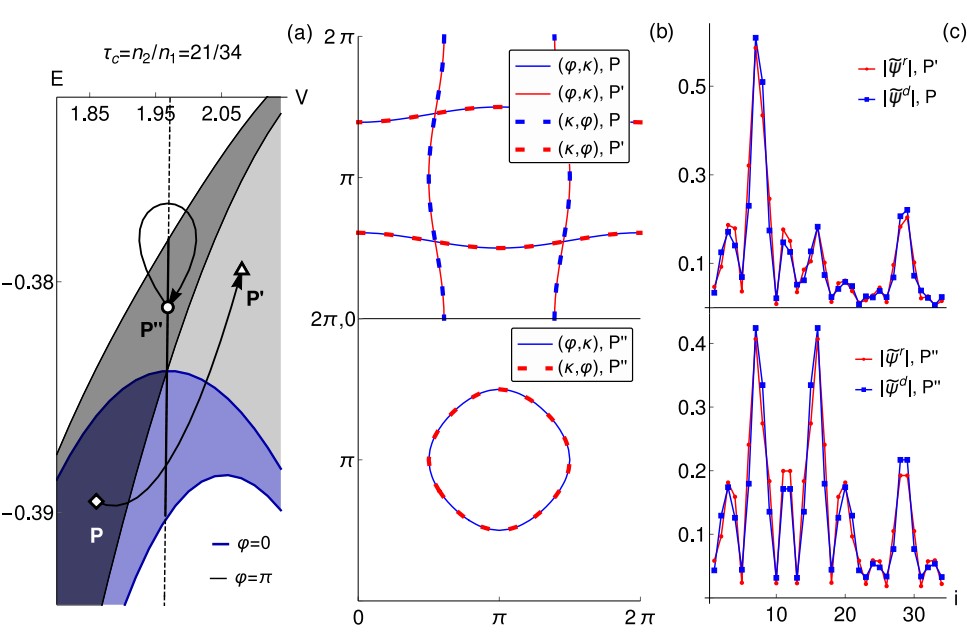

Figure 9: (a) Example of dual points for a randomly selected energy band of the CA defined by $\tau_c = 21/34$, for the AA-NNN, defined in Eq. 9 and below it, with next-nearest-neighbor hoppings $t_2 = 0.2$. (b) Top: FS at P (blue) and P' (red) and their $\pi/2$-rotation around $(\varphi_0, \kappa_0) = (\pi, \pi)$, respectively dashed blue and dashed red. Bottom: FS for the SD point $P''$ (blue) and its rotation (dashed red). (c) Top: Wave function $\tilde{\psi}^{\mathrm{r}}$ at $P'$ and its Aubry-André dual $\tilde{\psi}^{\mathrm{d}}$ at $P$. Bottom: Wave function $\tilde{\psi}^{\mathrm{r}}$ and its Aubry-André dual $\tilde{\psi}^{\mathrm{d}}$ at $P''$.

---

[3]Note that we should specify both the point $P(\lambda, E)$ and phases $\phi$ and $k$. This is because in general there is a subset of infinitely many phases $(\phi, k)$ associated with point $P$ (the FS is a line and not a point, in general). Even though two different phases in this subset correspond to the same point $P$, they may correspond to different wave functions.

As this mapping respects cyclical translations, $\mathcal{O}_c$ obeys

$$\mathcal{O}_c\left[\mathcal{T}^n\tilde{\psi}^{\mathrm{d}}(P;\phi,k)\right]=\mathcal{T}^n\tilde{\psi}^{\mathrm{r}}(P';\mathcal{R}_0[\phi,k]^T),\quad n=0,\cdots,n_1-1\,,\tag{33}$$

where $\mathcal{T}$ is the cyclic translation operator previously defined [4]. This implies $\mathcal{O}_c$ is a circulant matrix, i.e. its rows are just cyclic translations of the first row. Therefore, since any circulant matrix is diagonalized by the discrete Fourier transform, $\mathcal{O}_c$ is entirely defined by its eigenvalues. In other words Eq. (33) determines the duality transformation $\mathcal{O}_c$ uniquely for a given approximant.

To summarize, the full duality transformation can be obtained through a two-step procedure:

1. Find dual points in the phase diagram $P\equiv P\big(\boldsymbol{\lambda},E(\boldsymbol{\lambda},\varphi,\kappa)\big)$ and $P'\equiv P'\big(\boldsymbol{\lambda}',E'(\boldsymbol{\lambda}',\mathcal{R}_0[\varphi,\kappa]^T)\big)$, associated with FS that are identical under rotation $\mathcal{R}_0$;

2. Find the duality matrix $\mathcal{O}_c$, defined in Eq. (33), that maps the wave functions $\tilde{\psi}^{\mathrm{d}}(P;\phi,k)$ at $P$ and $\tilde{\psi}^{\mathrm{r}}(P';\mathcal{R}_0[\phi,k]^T)$ at $P'$ and their cyclic translations.

In Appendix B, we describe an efficient method to carry out step 1. Step 2 will be detailed with examples in the following sections. Schematically, considering that the Hamiltonian depends on a set of parameters $\boldsymbol{\lambda}$, the full duality transformation for a given energy band reads

$$
\begin{array}{ccc}
& \text{Full duality} & \\
P\big(\boldsymbol{\lambda},E(\boldsymbol{\lambda},\varphi,\kappa)\big) & \longleftrightarrow & P'\big(\boldsymbol{\lambda}',E'(\boldsymbol{\lambda}',\mathcal{R}_0[\varphi,\kappa]^T)\big) \\
\downarrow & & \downarrow \\
& \mathcal{O}_c & \\
\mathcal{T}^j\tilde{\psi}^{\mathrm{d}}(P;\phi,k) & \longleftrightarrow & \mathcal{T}^j\tilde{\psi}^{\mathrm{r}}(P';\mathcal{R}_0[\phi,k]^T) \\
j=0,\cdots,n_1-1 & & j=0,\cdots,n_1-1
\end{array}\;.\tag{34}
$$

This duality is illustrated in Fig. 10. The duality is only local, in the sense that it is defined for each energy band of a CA, in the neighborhood of the critical (self-dual) point. In general, these *local dualities* may break-down sufficiently away from SD points, i.e. there might be no pair $\boldsymbol{\lambda}$ and $\boldsymbol{\lambda}'$ associated with FS identical under any rotation $\mathcal{R}_0$. Nevertheless, for a the generic families we studied, there is always a set of dual points in the vicinity of a self-dual point.

**Testing the generalized duality.—** As noted in the previous section, for the AAM, the duality matrix $\mathcal{O}_c$ is always the identity because $\tilde{\psi}^{\mathrm{d}}(P;\phi,k)=\tilde{\psi}^{\mathrm{r}}(P';\mathcal{R}_0[\phi,k]^T)$ for any CA. In order to test the validity of $\mathcal{O}_c$ in a non-trivial example, we consider the MAAM, for which an analytical expression for the duality transformation was found in Ref. [34]. Note however that this transformation is not unique at the self-dual points . In fact, we found that, defining the eigenstates for the MAAM (particular case of Eq. 9) as $|\psi\rangle=\sum_n\psi_n c_n^\dagger|0\rangle$ in the infinite-size incommensurate limit (for irrational $\tau$), the wave function is also self-dual under the transformation (see Appendix D)

$$\psi_l'=\sum_n e^{-2\pi i\tau ln}\chi_n(\beta_0,\tau)\psi_n\,,\tag{35}$$

where $\psi_l'$ is the dual wave function (not to confuse with the Aubry-André dual wave function $\psi_l^d=\sum_n e^{-2\pi i\tau ln}\psi_n$), the parameters of the model are $\boldsymbol{\lambda}=\{t,V,\alpha\}$,

---

[4]We assume the matrix $\mathcal{Q}_d$ containing $\mathcal{T}^n\tilde{\psi}(P;\phi,k)$, $n=0,\cdots,n_1-1$ in its columns is not rank-deficient, that is rank($\mathcal{Q}_d$) $=n_1$. However, one should check whether this is true for the states being used before computing $\mathcal{O}_c$, otherwise the latter is not well-defined.

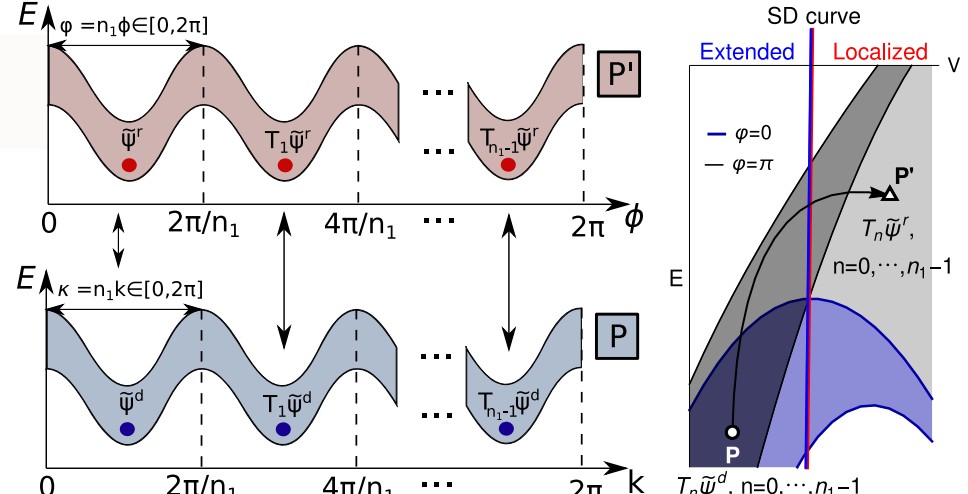

Figure 10: Pictorial description of the duality transformation. The energy bands are periodic in $\phi$ and $k$ with period $2\pi/n_1$. However, the wave function differs by cyclic translations in each of these periods. The different $\phi$-periods ($k$-periods) contain all the $n_1 - 1$ possible cyclic translations of the first-period real-space wave function $\tilde{\psi}^{\mathrm{r}}$ (dual wave function $\tilde{\psi}^{\mathrm{d}}$). If $P$ and $P'$ are dual points of a given CA, $\mathcal{O}_c$ is the transformation that transforms $T_n \tilde{\psi}^{\mathrm{d}}$ at $P$ into $T_n \tilde{\psi}^{\mathrm{r}}$ at $P'$, with $n = 0, \cdots, n_1 - 1$.

$\chi_n(\beta_0, \tau) = \sinh\beta_0 [\cosh\beta_0 - \cos(2\pi\tau n)]^{-1}$, with $\beta_0$ defined as $2t\cosh\beta_0 = E + 2V\alpha^{-1}$. For a given CA defined by $\tau_c$, we define

$$\tilde{\psi}'_p = \sum_{m=0}^{n_1-1} e^{-2\pi i \tau_c p m} \chi_c^m \tilde{\psi}^{\mathrm{r}}_m, \tag{36}$$

where $\chi_c^m \equiv \chi_m(\beta_0, \tau_c)$. This transformation reduces to Eq. (35) when $n_1 \to \infty$ ($\tau_c \to \tau$). Note that we used the prime notation in $\tilde{\psi}'_p$ to not confuse it with $\tilde{\psi}^{\mathrm{d}}_p$, the Aubry-André dual wave function defined in Eq. 32. Using the vector notation, it can also be written as

$$\tilde{\boldsymbol{\psi}}' = U^\dagger \boldsymbol{\chi}_c \tilde{\boldsymbol{\psi}}^{\mathrm{r}} = (U^\dagger \boldsymbol{\chi}_c U) U^\dagger \tilde{\boldsymbol{\psi}}^{\mathrm{r}} = (U^\dagger \boldsymbol{\chi}_c U) \tilde{\boldsymbol{\psi}}^{\mathrm{d}}, \tag{37}$$

where $\tilde{\boldsymbol{\psi}}' = \left\{ \tilde{\psi}'_0, \cdots, \tilde{\psi}'_{n_1-1} \right\}^T$, $U$ and $\boldsymbol{\chi}_c$ are matrices with entries, respectively, $U_{np} = e^{2\pi i \frac{n_2}{n_1} np}$ and $\chi_c^{np} = \chi_c^p \delta_{np}$, with $n, p = 0, \cdots, n_1 - 1$, and we used that $U^\dagger \tilde{\boldsymbol{\psi}}^{\mathrm{r}} = \tilde{\boldsymbol{\psi}}^{\mathrm{d}}$. At SD points, the transformation in Eq. (37) maps the wave function into itself, up to a normalization, and therefore $\tilde{\boldsymbol{\psi}}' \propto \tilde{\boldsymbol{\psi}}^{\mathrm{r}}$. Assuming that $\tilde{\boldsymbol{\psi}}^{\mathrm{r}}$ and $\tilde{\boldsymbol{\psi}}'$ are normalized , we have that, at SD points:

$$\mathcal{O}_c = U^\dagger \boldsymbol{\chi}_c U. \tag{38}$$

We may now check if our definition for $\mathcal{O}_c$ in Eq. (33) matches the one obtained above. Since $\mathcal{O}_c$ is a circulant matrix, it is diagonalized by the unitary transformation $U$. Therefore, we just need to compute the eigenvalues of $\mathcal{O}_c$ and check if they match the values $\chi_c^p$. Note that if we define the function $\chi(\beta_0, x) = \frac{\sinh\beta_0}{\cosh\beta_0 - \cos(2\pi x)}$, $\chi_c^p$ are just evaluations of this function at points $x_p = \mathrm{mod}(n_2 p/n_1, 1)$. For a CA with $n_1$ sites in the unit cell, we sample $n_1$ points of function $\chi(x, \beta_0)$. The results are shown in Fig. 11 for the model in Eq. 6 of [34]. We can see that the eigenvalues of the computed matrices $\mathcal{O}_c$ perfectly fall on top of the $\chi(\beta_0, x)$ curve, after a global rescaling. We have computed $\mathcal{O}_c$ for fixed $\cosh\beta = 4$, using $\tilde{\boldsymbol{\psi}}^{\mathrm{r}}$ and $\tilde{\boldsymbol{\psi}}^{\mathrm{d}}$ at multiple SD points of a given CA, with $k = \phi = 0$ (note that at SD points, $\beta_0 = \beta$). The obtained eigenvalues shown in Fig. 11 were always the same, up to normalization. This was

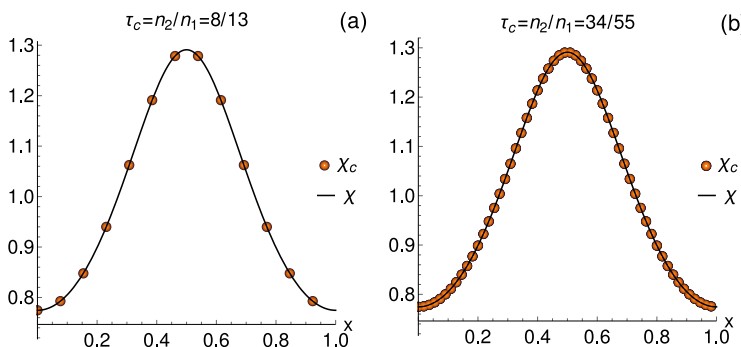

Figure 11: Results for the MAAM, defined in Eq. 9 and below it. Eigenvalues of matrices $\mathcal{O}_c$ (Eq. 38) defined for different energy bands of CA with $\tau_c = 8/13$ (a) and $\tau_c = 34/55$, for $\cosh\beta = 4$ and $\phi = k = 0$.(b). A total of 7 and 28 energy bands (points) were used respectively in (a) and (b). Different points have different sizes and colors, perfectly overlapping. The eigenvalues are compared with the function $\chi(\beta_0, x)$ defined in the text below Eq. 38. They were rescaled so that the eigenvalue corresponding to $x_p = 0$ matches the value $\chi(\beta_0, 0)$.

expected as for fixed $\beta$, $\chi_c^p$ does not depend on energy nor on the rest of the Hamiltonian's parameters.

**Application to generic models.—** The definition of $\chi_c$ can be easily generalized to other models using that the duality transformation, $\mathcal{O}_c$, defined in Eq. (33), is a circulant matrix and thus can always be written in the form of Eq. (38). Two dual points $P$ and $P'$ of a given approximant, $\tau_c = n_2/n_1$, of a quasiperiodic model define a set of eigenvalues $\chi_c^p$, which can be parametrized as the function $\chi_c(x)$ evaluated at $x = x_p = \mod(n_2 p/n_1, 1)$. In the following, we provide numerical evidence that the function $\chi_c$ can converge very fast with the order of the approximant, i.e. $\chi_c(x) \simeq \chi(x)$. Thus, the method we describe above to explicitly compute $\mathcal{O}_c$ provides a way of effectively approximating $\chi$ for generic models.

To illustrate our findings, we show in Fig. 12 the function $\chi_c(x)$ obtained for the AA-NNN and the LM. The SD points are indicated in the phase diagrams in Figs. 12(a,d). The CA's states used to define $\mathcal{O}_c$ were selected so that their energies were the closest possible to the selected points in the phase diagram. For the LM, $\mathcal{O}_c$ was defined from the real-space and dual wave functions and their translations within the upper layer (containing $n_1$ sites by definition). A similar duality transformation could be defined for the lower layer, from the real-space and dual wave functions within this layer.

Figs. 12(b,e) show that $\chi_c(x)$ is well-behaved in the sense that interpolations of the points $\chi_c(x_p)$ are essentially independent of the approximant. This means that the thermodynamic-limit function $\chi(x)$ can be obtained even for low-order CA. However, such clean behaviour only occurs for fine-tuned states, such as the highest-energy states marked with points $P_A$ and $P_C$ in Figs. 12(a,d). For other critical states, $\chi_c(x)$ expresses a more irregular behaviour, as shown in the examples of Figs. 12(c,f). In this case, $\chi_c(x)$ has singularity-like features that may only be resolved for large-order CA. The non-triviality of $\chi_c(x)$ in these cases shows that finding analytical descriptions of duality transformations in certain models can be challenging.

It is worth reinforcing that the duality transformation defined above is not restricted to SD points, and can be defined for any dual points $P$ and $P'$. An example of $\chi_c(x)$ computed at both SD and non-SD points is given in Fig. 13. Note that $\chi_c(x)$ changes smoothly as we move away from the critical point.

As a final remark, we note that we used here examples of duality transformations for which $(\phi_0, k_0) = (0, 0)$. In general, $(\phi_0, k_0)$ can take other values and that is observed for

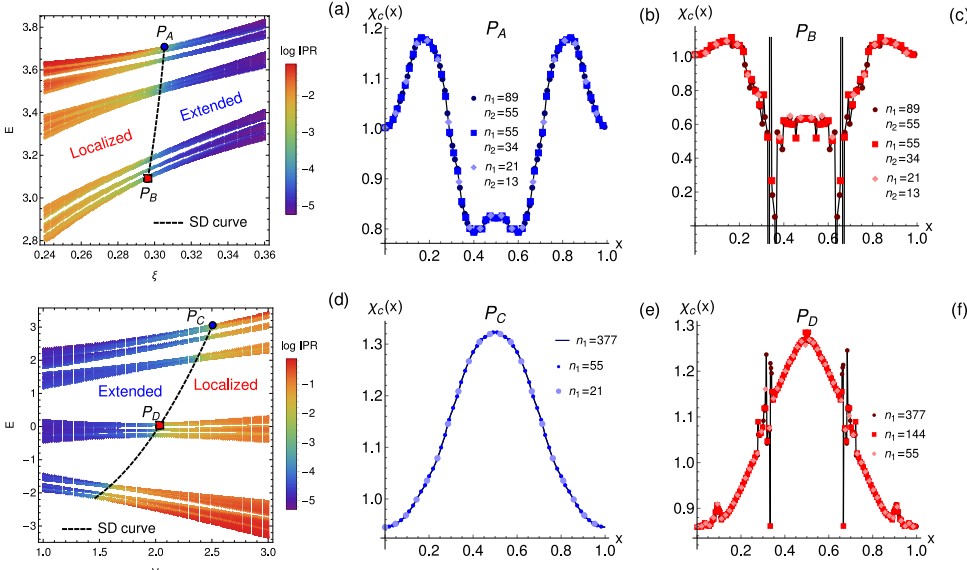

Figure 12: (a) IPR for the LM, defined in Eq. 10, with $V = 3, \Lambda = 2.5a$, for $\tau_i = 144/233$ [highest energy states in Fig. 2(c)]. The dashed black line corresponds to interpolated SD curves, computed for $\tau_c = 34/55$. (b,c) Eigenvalues of matrices $\mathcal{O}_c$ defined in Eq. 38, labeled as $\chi_c(x_p) \equiv \chi_c^p$, computed for different CA, for points in the SD curve marked in (a). (d) IPR for the AA-NNN, defined in Eq. 9, with $t_2 = 0.1$, for $\tau_i = 144/233$ (system with 233 sites). The SD curves were computed for $\tau_c = 34/55$. (b,c) $\chi_c(x_p)$ computed for the points marked in (d). In (c,f), all the points were connected with a black line to guide the eye. Function $\chi_c$ was computed for states with $\phi = k = 0$.

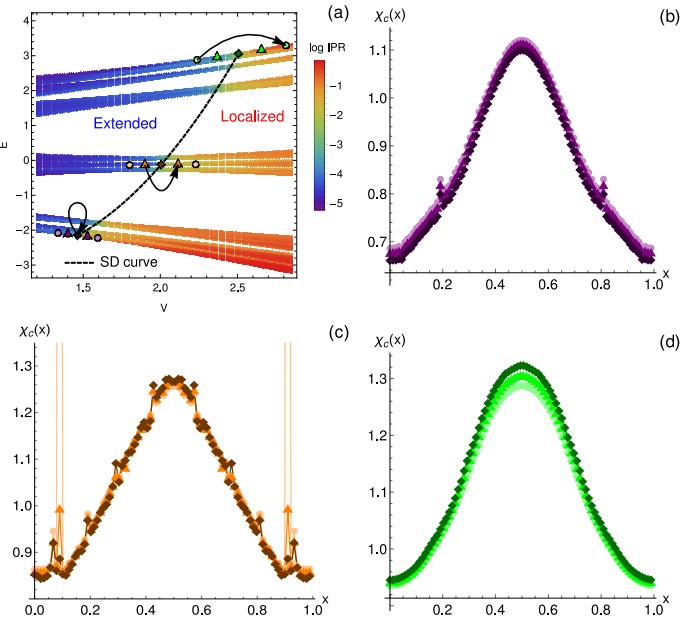

Figure 13: (a) IPR for the AA-NNN, defined in Eq. 9, with $t_2 = 0.1$ and selected dual points. These include points in the SD curve, that are self-dual, and non-self-dual dual points in the extended and localized phases represented by the same marker. (b-d) Eigenvalues of matrices $\mathcal{O}_c$ defined in Eq. 38, labeled as $\chi_c(x_p)$, computed for the dual points in (a), for the CA with $\tau_c = 55/89$. The points $\chi_c(x_p)$ in (b-d) are plotted with the markers of the corresponding dual points in (a).

other regions in the phase diagram, as for the AA-NNN and the LM. Typically, branches of the phase diagram that share continuously connected mobility edges (SD curves) share the same $(\phi_0, k_0)$. For instance, in the example of Fig. 2(d), the energy bands that cross the upper branch with positive concavity have $(\phi_0, k_0) = (0, 0)$, while for the lower branch with negative concavity we have $(\phi_0, k_0) = (\pi(n_1 + n_2)/n_1, 0)$, see Appendix. E.

# 7  Discussion

We have shown several examples of 1D QPS for which transitions between extended and localized phases are associated with local hidden dualities. We could not find any instance where such hidden duality was not present, at least sufficiently close to localization-delocalization critical point. Therefore, we conjecture that such dualities exist generically in 1D QPS and provide further arguments in the following.

The existence of dualities is deeply connected with the invariance of QPS under $\varphi$ and $\kappa$-shifts that have a period inversely proportional to the unit cell's size. In generic QPS, this invariance may be more complicated than in the previous examples. For instance, for EAAM, besides the on-site quasiperiodic energies, we could have other inhomogeneities between different sites. For concreteness, we consider $N$ consecutive sites that have different on-site energies, in addition to the quasiperiodic potential ($N$ should be bounded, otherwise the system would be disordered). In that case, the $\phi$-periodicity for a CA would change from $2\pi/n_1$ to $2\pi N/n_1$. For the latter, we should define $\varphi = n_1 \phi/N$, so that $\Delta\varphi = 2\pi$ contains a period of the energy dispersion. Obviously, the unit cell of the lowest-order CA is constrained to have, at least, $n_1 = N$ sites.

We tested this scenario in two different ways. The first was a simple example of the AAM with an additional staggered potential between any two consecutive sites ($\eta$-AAM, see Appendix C). In this case, after defining $\varphi = n_1 \phi/2$, we observe that even though no signs of duality exist for $\tau_c = 1/2$ (the lowest-order possible CA), $\tau_c = 21/34$ already hosts SD points for which the FS is invariant under interchanging the new $\varphi$ with $\kappa$.

We studied a second example of an AAM with additional different on-site energies for any three consecutive sites (3ICS-AAM, see Appendix C). In this case, $\varphi = n_1 \phi/3$ and there is an important qualitative difference with respect to the other examples that we studied. In particular, the center of rotation $(\varphi_0, \kappa_0)$ depends smoothly on energy and on the Hamiltonian's parameters. Remarkably, besides this difference, the FS for a CA of high-enough order around criticality also has the universal behavior shown in Fig. 6(b), being perfectly described locally by the model in Eq. (12). The SD points again perfectly matched the mobility edge of the limiting QPS.

All the results shown so far are in favour of the two main ideas of this work: (i) CA of generic QPS share the Aubry-André FS universality around criticality; (ii) Transitions between extended and localized phases in 1D QPS are associated with hidden dualities that manifest around criticality.

Regarding (i), we conjecture that the FS universality of CA around criticality is connected to an existing universality in localization-delocalization transitions in 1D QPS. For a given CA, the "critical" FS of Fig. 6(b) (dispersive both in $\kappa$ and $\varphi$) occur only within a narrow region of parameters around criticality. This region shrinks as the unit cell is increased, eventually collapsing to the QPS's critical points.

The critical universal FS that we observe for large unit cells are compatible with the vanishing of all the harmonics in $\varphi$ and $\kappa$ other than the fundamental in this limit (so that Eq. (12) is valid). The implications of the vanishing of non-fundamental harmonics near criticality for generic 1D QPS will be discussed elsewhere [54].

The hidden dualities mentioned in (ii) are deeply connected with the symmetry of CA to displacements encoded in phase $\phi$. This is apparent in our definition of $\mathcal{O}_c$ in Eq. (33). Such symmetry exists for CA of generic QPS, therefore the presence of hidden dualities is expected to arise for a much larger set of systems than the ones studied in this work. Our findings open a new route for the study and understanding 1D QPS. In particular, they provide a working criterion for the existence of mobility edges and a way of generating models of QPS with analytical phase diagrams by explicitly creating dualities in their simplest CA [54].

The scope of this work was to characterize quasiperiodicity-driven localization-delocalization transitions for which the FS for different CA can be characterized by the effective model in Eq. 12, at least close enough to the transition and for a large enough unit cell. Such transitions are associated with hidden dualities manifested in $\varphi$ and $\kappa$, that can be clearly seen through this effective model. However, in some cases, the effective model can be more general. This is the case for models that have phases with critical states over a finite range of parameters (and not only at fine-tuned points as at the critical points of localization-delocalization transitions). In this case, hidden dualities can also arise and may still be captured analytically in some models. We will cover this case in detail elsewhere [54]. On the other hand, there may be some models for which no localization-delocalization transition exists for any finite quasiperiodicity and therefore no hidden duality can be defined. An example is the Maryland model that we discuss in Appendix F, for which all the eigenstates are localized for any strength of the quasiperiodic potential. In this case, the effective model is given in terms of a $\tan(\varphi/2)$ term instead of $\cos(\varphi)$. Since there is no $\kappa$-dependent term of the type $\tan(\kappa/2)$, it is clear that no hidden duality exists in this case. Nonetheless, we can still see that the renormalized coupling that multiplies the $\kappa$-dependent term becomes irrelevant with respect to the $\varphi$-dependent term as the unit cell is increased for any finite $V$. This is in accordance with our view of the localized regime, where only the latter coupling should survive in the limit of large unit cell.

A natural next step is to understand if our ideas extend to more complex 1D systems and to higher dimensions where more exotic localization phenomena arise [21,43,44]. More complex 1D systems may include (i) models with internal degrees of freedom, for which the unit cell contains more than one site even in the absence of the quasiperiodic term in the Hamiltonian; and (ii) models with multiple quasiperiodic potentials [5]. Our results for the $\eta$-AAM and the 3ICS-AAM, examples of type-(i) models, suggest that hidden dualities are also present for models of this type close to localization-delocalization transitions. Another interesting open question is the influence of interactions on generalized dualities. Mobility edges for interacting systems were found to depart from the single-particle description in some regimes [51, 58]. Thus a natural question is whether these many-body mobility edges are also associated with hidden dualities that depart from CA.

# Acknowledgments

The authors acknowledge partial support from Fundação para a Ciência e Tecnologia (Portugal) through Grant and UID/CTM/04540/2019. BA and EVC acknowledge partial support from FCT-Portugal through Grant No. UIDB/04650/2020. MG acknowledges further support through the Grant SFRH/BD/145152/2019. BA acknowledges further support from FCT-Portugal through Grant No. CEECIND/02936/2017. We finally acknowledge the Tianhe-2JK

---

[5] A simple example is a model with two quasiperiodic potentials characterized by different irrationals $\tau_1 \neq \tau_2$ (BCM). Interestingly, CA of the $\eta$-AAM are also CA of the BCM, with $\tau_1 = \tau$ and $\tau_2$ an irrational close to 1/2. The same holds for the 3ICS-AAM, but in that case, $\tau_2$ can be an irrational close to 1/3 or 2/3. Therefore, the existence of hidden dualities in the $\eta$-AAM and 3ICS-AAM suggests that they are also present in CA of the BCM.

cluster at the Beijing Computational Science Research Center (CSRC), the Baltasar-Sete-Sóis cluster, supported by V. Cardoso's H2020 ERC Consolidator Grant no. MaGRaTh-646597, and the OBLIVION supercomputer (based at the High Performance Computing Center - University of Évora) funded by the ENGAGE SKA Research Infrastructure (reference POCI-01-0145-FEDER-022217 - COMPETE 2020 and the Foundation for Science and Technology, Portugal) and by the BigData@UE project (reference ALT20-03-0246-FEDER-000033 - FEDER and the Alentejo 2020 Regional Operational Program. Computer assistance was provided by CSRC, CENTRA/IST and the OBLIVION support team.

## A    Scaling analysis

In this section, we show that it is possible to carry out a scaling analysis in terms of the $\varphi$ and $\kappa$-dependent energy dispersions, $\Delta E_\varphi(\kappa^*) = |E_{\varphi=\pi}(\kappa^*) - E_{\varphi=0}(\kappa^*)|$ and $\Delta E_\kappa(\varphi^*) = |E_{\kappa=\pi}(\varphi^*) - E_{\kappa=0}(\varphi^*)|$. The aim is to inspect how these dispersions change upon increasing the order of the approximant.

To carry out the scaling analysis, it is important to study different CA at the same point in the phase diagram, which may be challenging. A possible way to do it is to recall that in the extended phase of the limiting QPS, for a CA of high-enough order, $E(\kappa, \varphi) \approx E(\kappa)$, while in the localized phase, $E(\kappa, \varphi) \approx E(\varphi)$. As seen in the main text, using larger approximants in such cases is similar to a band folding, in the $\kappa$ or $\varphi$ direction. We can fix a point $(\lambda, E)$ starting deep in the extended phase and, considering that $E(\kappa, \varphi) \approx E(\kappa)$, compute the wave vectors corresponding to such point for different approximants, $\kappa^*(\lambda, E, \tau_c)$. We can then see how states with fixed $\kappa^*$ evolve with $\lambda$ for different approximants. As long as we are sufficiently away from criticality or using a CA of high-enough order, if there is a state at point $(\lambda, E)$, there will also be a state at the same point for a higher-oder CA. In that case, a finite-size scaling analysis can be carried out at point $(\lambda, E)$, by increasing the size of the unit cell (the CA's order).

As an example of application, we use the AA-NNN with $t_2 = 0.5$. In Fig.14(a), we can see that the points of fixed $\kappa^*(V, E, \tau_c)$ in the figure change with $V$, starting at the smallest-$V$

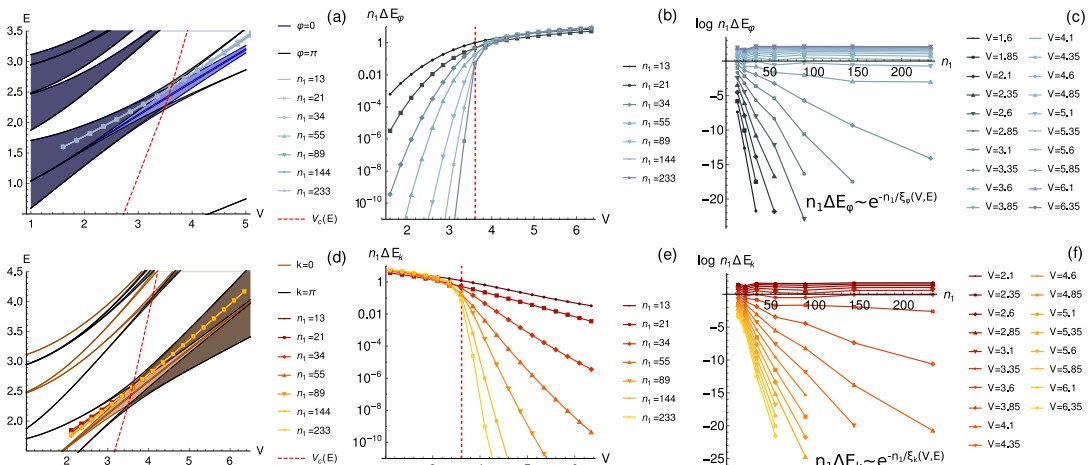

Figure 14: Examples of $\Delta E_\varphi \equiv \Delta E_\varphi(\kappa^* = 0)$ and $\Delta E_\kappa \equiv \Delta E_\kappa(\varphi^* = 0)$ for the AA-NNN with $t_2 = 0.5$. These quantities were computed for the points marked in (a,b). In (b,e) we plot fixed-$n_1$ curves for variable $V$ and in (c,f) we do the opposite. The energy bands shown in (a,b) are for $\tau_c = 8/13$. See text for a more detailed explanation.

point. These points fall on top of each other for different approximants as a consequence of the arguments given above. The same can be done in the localized phase, but there one must compute the phase $\varphi^*(V, E, \tau_c)$ for a point $(V, E)$ deep in the localized phase, and then follow this state upon varying $V$. An example of application is in Fig. 14(d).

We can finally compute $\Delta E_\varphi \equiv \Delta E_\varphi(\kappa^* = 0)$ and $\Delta E_\kappa \equiv \Delta E_\kappa(\varphi^* = 0)$ for different $(V, E)$ points, which are exponentially small respectively in the extended and localized phases. $\Delta E_\varphi$ is shown in Figs. 14(b,c) for the points represented in Fig. 14(a) and $\Delta E_\kappa$ is in Figs. 14(e,f) for the points represented in Fig. 14(d). In the extended phase, we have $\Delta E_\kappa \sim n_1^{-1}$, while in the localized phase, $\Delta E_\varphi \sim n_1^{-1}$. This is because the bands of higher-order CA are, approximately, just being folded. Therefore, in Figs. 14(b,c,e,f) we plot the quantities $n_1 \Delta E_\kappa$ and $n_1 \Delta E_\varphi$. In Figs. 14(c,f), we see that in the localized phase, $n_1 \Delta E_\kappa \sim e^{-n_1/\xi_\kappa(V, E)}$ and in the extended phase, $n_1 \Delta E_\varphi \sim e^{-n_1/\xi_\varphi(V, E)}$. These exponential decays define the correlation lengths $\xi_\kappa$ and $\xi_\varphi$, respectively in the localized and extended phases. Such length scales can be extracted by fitting the data in Figs. 14(c,f).

By considering other paths similar to the ones in Figs. 14(a,d), we can compute the correlation lengths at different points in the phase diagram. An example of such computation is shown Fig. 15. There, we see that $\xi_\kappa$ and $\xi_\varphi$ diverge at the critical point. In Fig. 15(b) we compare the correlation lengths with the inverse participation ratio (IPR) and the momentum-space inverse participation ratio (IPR$_\kappa$). The former is defined in the main text, while the latter is defined as IPR$_\kappa(E) = (\sum_n |\psi_n^\kappa(E)|^2)^{-2} \sum_n |\psi_n^\kappa(E)|^4$, where $\psi_n^\kappa(E)$ is the Fourier-transform

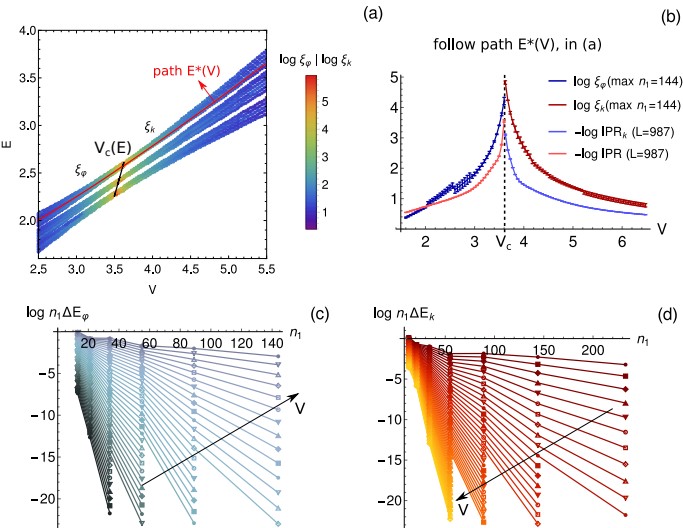

Figure 15: (a) Density plot of $\log \xi_\varphi$ in the localized phase, and $\log \xi_\kappa$ in the extended phase. This example is for the the AA-NNN with $t_2 = 0.5$. The full black line indicates the mobility edge. (b) $\log \xi_\varphi$ and $\log \xi_\kappa$ computed for the path shown in (a) and comparison with the IPR and IPR$_\kappa$ along the same path, for a system with $L = 987$ sites. (c,d) $\log n_1 \Delta E_\varphi$ and $\log n_1 \Delta E_\kappa$ as a function of the unit cell's size, $n_1$. $V$ increases in the direction of the arrow. Note that the apparent discontinuities in (b) for $\xi_\kappa$ and $\xi_\varphi$ can be understood in (c/d): for a CA of high-enough order, $\Delta E_\varphi$ and $\Delta E_\kappa$ fall below machine precision and such data points cannot be considered in the fit to extract $\xi_\kappa$ and $\xi_\varphi$. Therefore, the fitting procedure involves different approximants, depending on the coupling $V$. For the fit, we considered the 3 approximants corresponding to the smallest $\Delta E_\varphi$ or $\Delta E_\kappa$. As we are not in the thermodynamic limit and there is still some dependence on the size of the unit cell, a change in the group of fitted approximants can translate in apparent slight discontinuities in $\xi_\kappa$ and $\xi_\varphi$.

of the real-space wave function. The IPR and IPR$_\kappa$ also diverge at the critical point and can define correlation lengths respectively in the localized and extended phase. $\xi_\kappa$ and $\xi_\varphi$ provide alternative definitions for the correlation lengths without the explicit knowledge of the wave function.

## B  Finding SD points and Mapping dual points in the phase diagram

We have seen in the main text that the knowledge of the CA's FS in the $(\varphi, \kappa)$ plane is a powerful way to identify dual points in the phase diagram, and, in particular, SD points. This can be done analytically for low-order approximants, but may become challenging for higher-order ones. In this section we provide methods to compute dual points for generic models and CA.

### B.1  Renormalized single-band model

For a CA of high-enough order and close enough to quasiperiodicity-driven localization-delocalization transitions, we have observed that the FS in the $(\varphi, \kappa)$ plane reduces to the sinusoids shown in Figs. 5 and 6, irrespectively of the model that we studied. Such FS can be simply described through the following expression for the energy dispersion around energy $E_R$:

$$E = E_R(\lambda, E) + V_R(\lambda, E) \cos[\varphi - \varphi_0(\lambda, E)] + 2t_R(\lambda, E) \cos[\kappa - \kappa_0(\lambda, E)],\qquad (39)$$

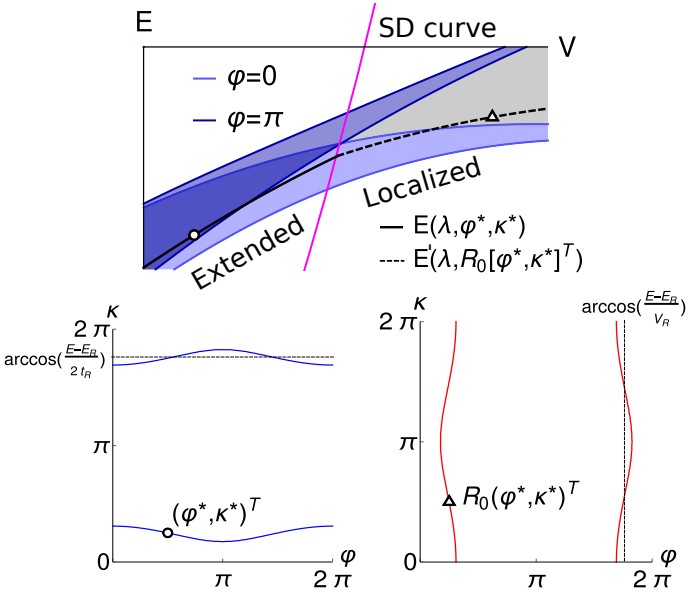

Figure 16: Sketch to illustrate how dual points can be found. For a given energy band, fixing the phases $\varphi^*$ and $\kappa^*$, we can define the curve $E(\lambda, \varphi^*, \kappa^*)$ in the extended phase (full black line) and the curve $E'(\lambda, \mathcal{R}_0[\varphi^*, \kappa^*]^T)$ in the localized phase (dashed line). These curves cross exactly at the SD curve (magenta). The FS associated with arbitrarily chosen dual points within these curves (circle and triangle) are shown in the bottom panel. Note that these dual points automatically satisfy Eqs. 42,43, with the latter being automatically fulfilled due to the choice of curves $E$ and $E'$.

where $V_R$ and $t_R$ are renormalized couplings, $(\varphi_0, \kappa_0)$ is the point around which the FS is invariant under a suitable $\varphi \leftrightarrow \kappa$ interchange at SD points, and $\boldsymbol{\lambda}$ are parameters of the Hamiltonian. This is the energy dispersion of a renormalized single-band AAM, being the generalization of the renormalized model defined in Ref. [37] (due to the energy-dependence of the renormalized couplings). It is important to let the renormalized couplings depend both on $\boldsymbol{\lambda}$ and $E$. If the sub-bands were perfect Aubry-André single-bands (from AAM with $\tau_c = 1$), there should be no energy dependence. However, in that case, there would be no mobility edges. To see this, just notice that, for fixed $E$:

$$\cos(\varphi - \varphi_0) = \frac{E - E_R}{V_R} - \frac{2t_R}{V_R} \cos(\kappa - \kappa_0), \tag{40}$$

$$\cos(\kappa - \kappa_0) = \frac{E - E_R}{2t_R} - \frac{V_R}{2t_R} \cos(\varphi - \varphi_0). \tag{41}$$

The generalized duality conditions that map points $(\boldsymbol{\lambda}, E)$ to dual points $(\boldsymbol{\lambda}', E')$ are

$$\frac{2t_R(\boldsymbol{\lambda}, E)}{V_R(\boldsymbol{\lambda}, E)} = \frac{V_R(\boldsymbol{\lambda}', E')}{2t_R(\boldsymbol{\lambda}', E')}, \tag{42}$$

$$\frac{E - E_R(\boldsymbol{\lambda}, E)}{V_R(\boldsymbol{\lambda}, E)} = \frac{E' - E_R(\boldsymbol{\lambda}', E')}{2t_R(\boldsymbol{\lambda}', E')}. \tag{43}$$

If $t_R$ and $V_R$ only depend on $\boldsymbol{\lambda}$, then the first equation fixes the transformation $\boldsymbol{\lambda}'(\boldsymbol{\lambda})$ which becomes energy independent. This is not what is observed in generic models.

Equations (42) and (43) allow us to find dual points in the phase diagram, including SD points $(\boldsymbol{\lambda}_c, E_c)$ that satisfy

$$V_R(\boldsymbol{\lambda}_c, E_c) = 2t_R(\boldsymbol{\lambda}_c, E_c). \tag{44}$$

In practice, if we compare curves $E(\boldsymbol{\lambda}(t), \varphi^*, \kappa^*)$ with curves $E'(\boldsymbol{\lambda}'(t), \mathcal{R}_0[\varphi^*, \kappa^*]^T)$, where $\boldsymbol{\lambda}(t)$ and $\boldsymbol{\lambda}'(t)$ are parametric curves in the space of parameters $\boldsymbol{\lambda}$ constructed by fixing the phases $\varphi^*, \kappa^*$, the condition in Eq. (43) is immediately satisfied for all points in those curves. Then we just need to identify dual points within these curves that satisfy Eq. (42), or in the case of SD points, Eq. (44). An illustration of the curves $E(\boldsymbol{\lambda})$ and $E'(\boldsymbol{\lambda}')$, along with the FS on both sides of the transition, is shown in Fig. 16. A detailed discussion on how to determine the renormalized couplings is given below.

## B.2 Estimation of the renormalized couplings

We here assume that point $(\varphi_0, \kappa_0)$ is known and use new coordinates $(\varphi, \kappa) \to (\varphi - \varphi_0, \kappa - \kappa_0)$ such that $(\varphi_0, \kappa_0)$ is shifted into the origin. For fixed $\boldsymbol{\lambda} = \boldsymbol{\lambda}^*$ and a given sub-band, we choose a grid of points $(\varphi, \kappa)$. Let $(\varphi^*, \kappa^*)$ be one of such points. It fixes an energy $E^* = E(\boldsymbol{\lambda}^*, \varphi^*, \kappa^*)$. For the phase diagram point $(\boldsymbol{\lambda}^*, E^*)$, we estimate the couplings as follows:

$$V_R = \frac{E(\varphi^* + \delta\varphi/2, \kappa^*) - E(\varphi^* - \delta\varphi/2, \kappa^*)}{\cos(\varphi^* + \delta\varphi/2) - \cos(\varphi^* - \delta\varphi/2)}, \tag{45}$$

$$t_R = \frac{1}{2} \frac{E(\varphi^*, \kappa^* + \delta\kappa/2) - E(\varphi^*, k^* - \delta\kappa/2)}{\cos(\kappa^* + \delta\kappa/2) - \cos(\kappa^* - \delta\kappa/2)}, \tag{46}$$

$$E_R = E(\varphi^*, \kappa^*) - 2t_R \cos(\kappa^*) - V_R \cos(\varphi^*), \tag{47}$$

where $\delta\kappa, \delta\varphi \ll 2\pi$. Note that for a perfect sinusoidal sub-band, these couplings are energy-independent [and therefore do not depend on the point $(\varphi^*, \kappa^*)$]. In general however, we will

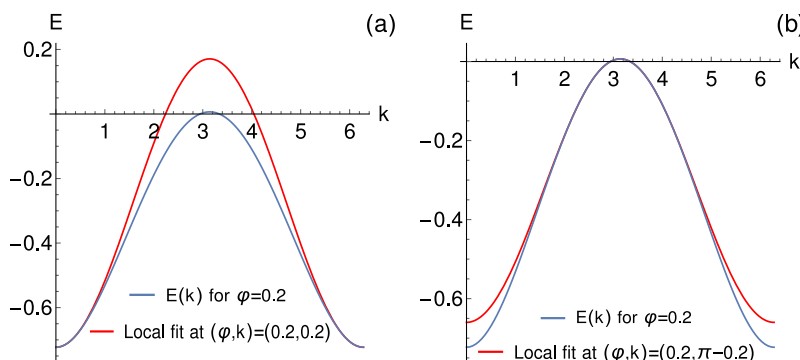

Figure 17: Examples of computation of the renormalized parameters $V_R, t_R$ and $E_R$. We locally estimate these parameters for a given sub-band, with the procedure introduced in Sec. B.2. In this figure we show examples of such estimation for $(\kappa, \varphi) = (0.2, 0.2)$ (a) and $(\kappa, \varphi) = (\pi - 0.2, 0.2)$, for $\delta \kappa = \delta \varphi = 0.15$. The obtained model (in red) is energy-dependent. These examples are for the AAM with $\tau_c = 2/3$, for the sub-band of intermediate energy.

not have perfect sinusoidal sub-bands, and the couplings become energy dependent as in the example shown in Fig. 17 [6].

In Figs. 18 and 19 we selected some example points in the phase diagram for the AA-NNN and the LM, and compared the exact FS with the one obtained through the renormalized single-

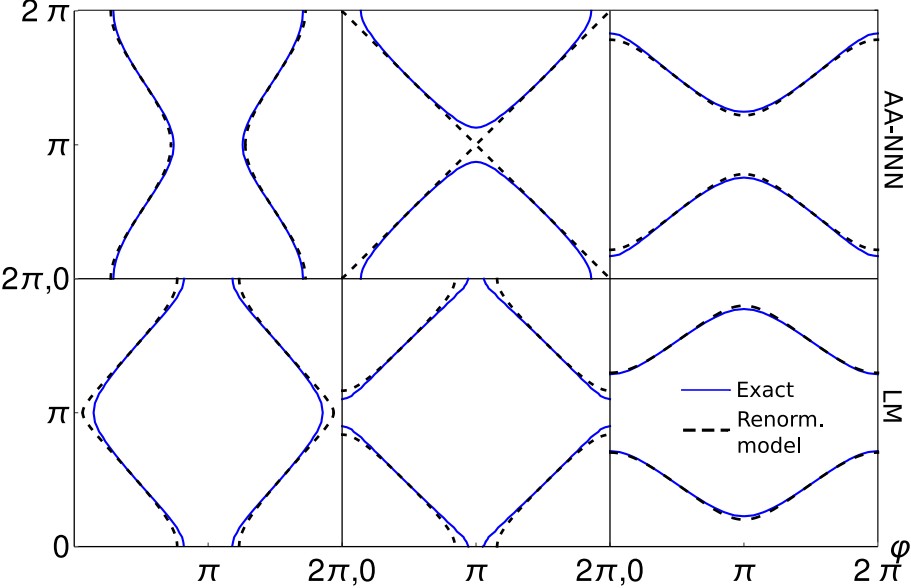

Figure 18: Examples of FS obtained for the CA defined by $\tau_c = 2/3$, for some selected points in the phase diagram. The exact result (full blue) is compared with the FS obtained with the single-band model in Eq. (39). The results in the upper row are for the AA-NNN, with $t_2 = 0.5$, lowest energy band and (from left to right) $V = 3.5; 2.76; 2.4$, $(\varphi^*, \kappa^*) = (\pi/2, \pi/2)$. The results in the lower row are for the LM, with $V = 3$, $\Lambda = 2.5$ and (from left to right) $\xi = 0.35; 0.365; 0.4$, $(\varphi^*, \kappa^*) = (\pi/2, \pi/2)$. We used $\delta \kappa = \delta \varphi = \pi/100$.

---

[6]It is important to be careful with points $P_{\varphi \kappa}$ around which $E(\kappa)$ is even [such as $(\varphi^*, \kappa^*) = (0, 0)$] otherwise the renormalized couplings as defined in Eqs. (45)-(47) will diverge. In such cases, we can just choose points $P_{\varphi \kappa} + (\epsilon, \epsilon)$, with small $\epsilon$.

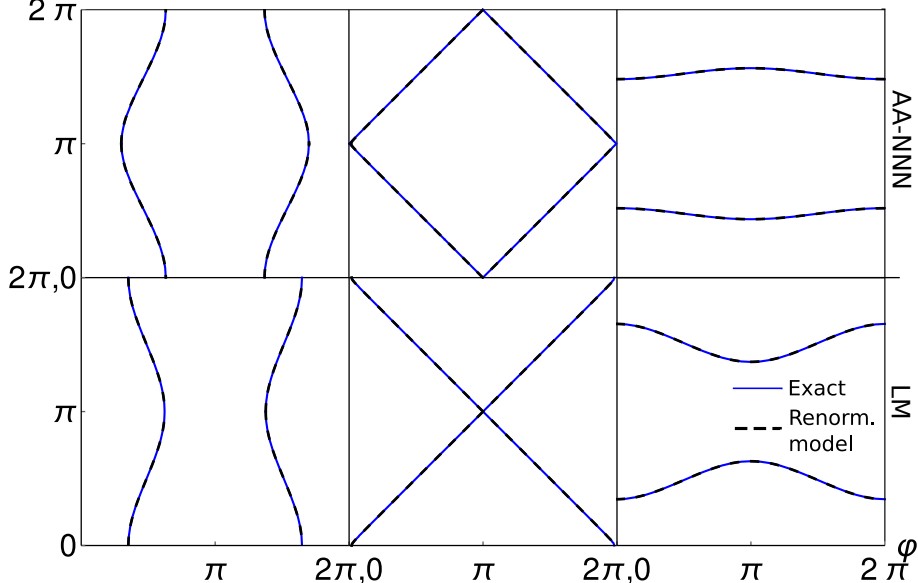

Figure 19: Examples of FS obtained for the CA defined by $\tau_c = 13/21$, for some selected points in the phase diagram. The exact result (full blue) is compared with the FS obtained with the single-band model in Eq. (39). The results in the upper row are for the AA-NNN, with $t_2 = 0.5$, $2^{\text{nd}}$ lowest energy band and (from left to right) $V = 2.2; 2.0915; 1.8$, $(\varphi^*, \kappa^*) = (\pi/2, \pi/2); (\pi, 0); (\pi/2, \pi/2)$. The results in the lower row are for the LM, with $V = 3$, $\Lambda = 2.5$ and (from left to right) $\xi = 0.35; 0.3655; 0.38$, $(\varphi^*, \kappa^*) = (\pi/2, \pi/2); (\pi, \pi); (\pi/2, \pi/2)$. We used $\delta\kappa = \delta\varphi = \pi/100$.

band model in Eq. (39). For $\tau_c = 2/3$, there are some deviations: the approximation is good close to the chosen point $(\varphi^*, \kappa^*) = (\pi/2, \pi/2)$, but fails away from it. For the higher-order approximant, $\tau_c = 13/21$, the agreement is perfect for the whole FS.

## B.3 Behaviour of renormalized couplings

Here we exemplify the behaviour of the renormalized couplings for the AAM. The idea of coupling renormalization has been previously discussed in Ref. [37].

We start by analyzing the examples provided in Fig. 20. We see that in the extended phase, we have $|V_R| < |2t_R|$ [Fig. 20(a)], while in the localized phase, we have $|V_R| > |2t_R|$ [Fig. 20(c)]. At the critical point, $V_R = 2t_R$ [Fig. 20(b)]. As the CA's order is increased, the ratio $2t_R/V_R$ increases in the extended phase [Fig. 20(d)] and decreases in the localized phase [Fig. 20(f)]. At the critical point, it remains unity [Fig. 20(e)]. Even though we used the AAM as an example, this is what occurs in generic models:

- The coupling $V_R$ ($t_R$) becomes irrelevant in the extended (localized) phase as $\tau_c \to \tau$;

- The couplings $V_R$ and $t_R$ are always relevant at the critical point.

We have seen in Sec. A that $\Delta E_\varphi \sim e^{-n_1/\xi_\varphi(V,E)}$ in the ballistic phase and $\Delta E_\kappa \sim e^{-n_1/\xi_\kappa(V,E)}$ in the localized phase. Therefore, we can also conclude that $V_R$ and $t_R$, which measure energy dispersion with $\varphi$ and $\kappa$, respectively, also behave as $V_R \sim e^{-n_1/\xi_\varphi(V,E)}$ in the extended phase and $t_R \sim e^{-n_1/\xi_\kappa(V,E)}$ in the localized phase.

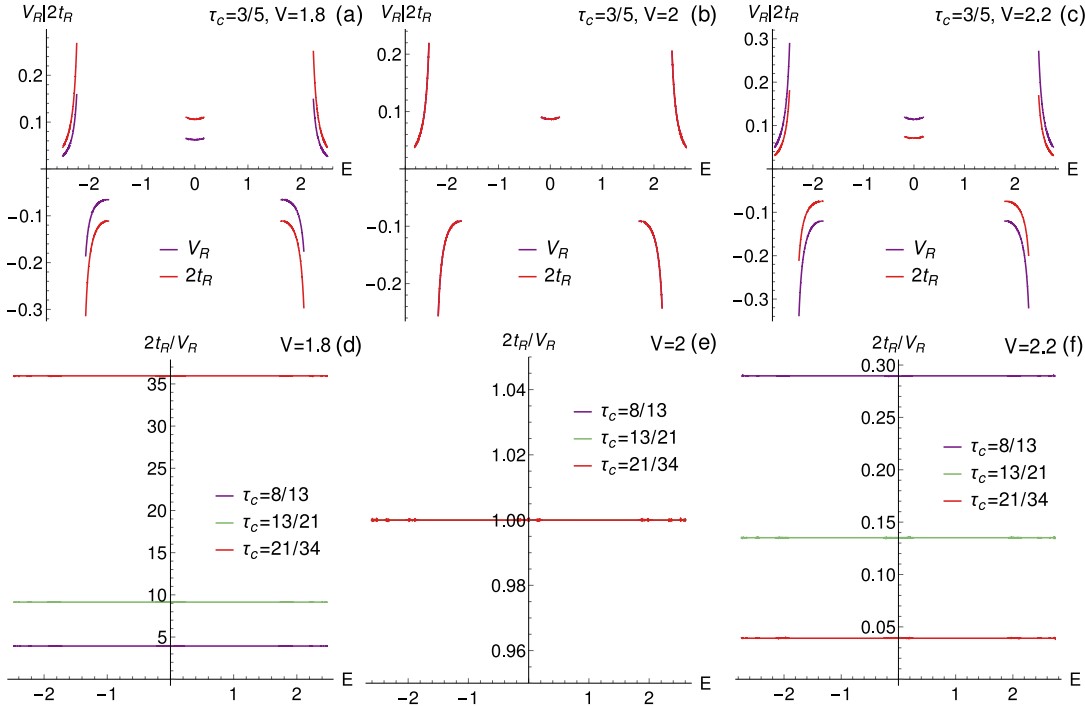

Figure 20: (a-c) Renormalized couplings $V_R$ and $2t_R$ for all the energy bands of the CA with $\tau_c = 3/5$, for $V = 1.8$ (extended phase), $V = 2$ (critical point) and $V = 2.2$ (localized phase). Note that the couplings are energy dependent which means that the sub-bands are not simple cosines. This is also the case for larger approximants. (d-f) Ratio $2t_R/V_R$ as a function of energy, for different approximants and $V$. This ratio is energy-independent as a consequence of the lack of a mobility edge and the energy-independent correlation lengths in the AAM [4]. For $V = 1.8$, the ratio increases with the CA's order, meaning that $V_R$ becomes irrelevant with respect to $t_R$, as expected in the extended phase. For $V = 2.2$, $2t_R/V_R$ decreases with the approximant - $t_R$ becomes irrelevant as expected in the localized phase. At the critical point, $V = 2$, the ratio is scale-invariant.

## B.4 Mapping dual points

We have already seen that the conditions in Eqs. (42) and (43) allow us to map dual points. Here we present examples of application for the AAM and the AA-NNN.

Dual points can be mapped with a two-step procedure:

1. Find curves $E(V)$ of constant $\chi_R = \frac{E - E_R}{V_R}$. Map them into curves $E'(V')$ of constant $\chi'_R = \frac{E' - E'_R}{2t'_R} = \chi_R$. This can be done by mapping curves $E(V, \varphi, \kappa)$ into curves $E'(V', \mathcal{R}_0[\varphi, \kappa]^T)$;

2. Map dual points within the curves above by considering $\Lambda_R \equiv 2t_R/V_R = V'_R/(2t'_R) \equiv \Lambda'_R$.

Defining the following quantities,

$$
\chi_R^f = \begin{cases} \chi_R \equiv \frac{E - E_R}{V_R}, & \chi_R \leq \chi'_R, \\ \chi'_R \equiv \frac{E' - E'_R}{2t'_R}, & \chi_R > \chi'_R, \end{cases} \tag{48}
$$

$$
\Lambda_R^f = \begin{cases} \Lambda_R \equiv 2t_R/V_R, & \Lambda_R \leq \Lambda'_R, \\ \Lambda'_R \equiv V'_R/(2t'_R), & \Lambda_R > \Lambda'_R, \end{cases} \tag{49}
$$

we can visually observe the duality between the extended and localized phases, given some energy band. Examples are in Fig. 21.

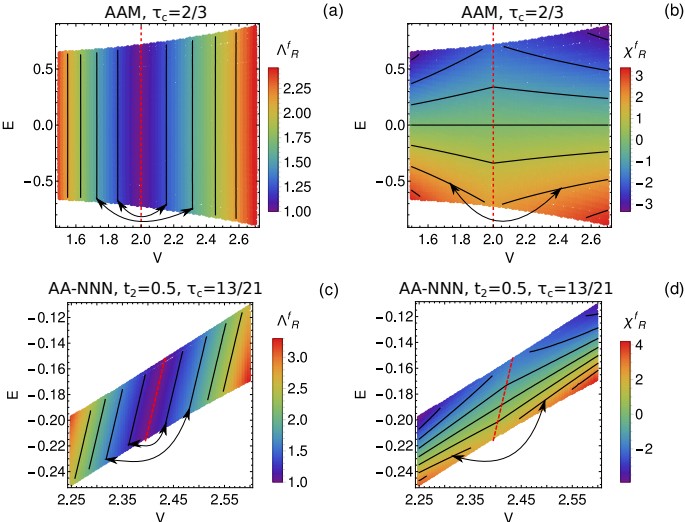

Figure 21: Quantities $\chi_R^f$ and $\Lambda_R^f$ defined in Eqs. (48) and (49) for the AAM with $\tau_c = 2/3$ (a,b) and the AA-NNN with $t_2 = 0.5$ and $\tau_c = 13/21$ (c,d), for selected energy bands. The black lines depict contours over which these quantities are constant. The red dashed lines depict the critical line, for which $\Lambda_R^f = 1$. The arrows relate examples of contours below and above the critical point with the same $\chi_R^f$ or $\Lambda_R^f$. Dual points share both the same $\chi_R^f$ and $\Lambda_R^f$.

## B.5 Absence of duality for low-order CA

As seen in the main text, for generic models a (almost) perfect FS duality only emerges for CA of high-enough order. For these models, the description of the sub-bands of a low-order CA in terms of Eq. (39) should fail, as already hinted in Fig. 18. For CAs of high-enough order, this simple description of the FS ensures the existence of dual points as long as the conditions in Eqs. (42) and (43) are satisfied.

Here we exemplify the breakdown of the renormalized single-band description for low-order approximants, in the AA-NNN model. In Fig. 22 we compute the ratio $2t_R/V_R$ for fixed $t_2 = 0.5$ and $V = 1.5$, for the whole spectrum and different approximants. There are two important comments:

- For smaller approximants, there is no well defined curve $\Lambda_R(E) \equiv 2t_R(E)/V_R(E)$. For a given energy, $\Lambda_R$ can take a finite range of values, which means that the description of the FS in terms of Eq. (39) is not suitable. Under the conjecture that the duality emerges when the energy dispersion becomes sinusoidal, no true duality exists for small approximants. It only emerges when we consider higher-order approximants for which the curve $\Lambda_R(E)$ becomes well-defined;

- The ratio $\Lambda_R(E)$ is energy-dependent for a fixed approximant, in contrast with the AAM. This follows from the existence of a mobility edge and energy-dependent correlation lengths.

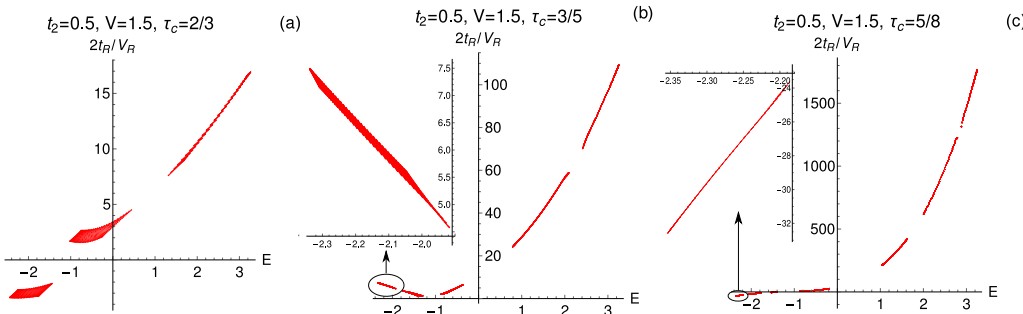

Figure 22: Ratio of renormalized couplings, $2t_R/V_R$, for fixed $t_2 = 0.5$ and $V = 1.5$ (AA-NNN model), for the whole spectrum of each CA. The CA's order is increased from (a) to (c). The renormalized parameters were computed for a grid of points $(\varphi, k)$, as explained in Sec. B.2. The insets show a close-up of the results for the lowest energy band to show that the values of $2t_R/V_R$ collapse to a well-defined curve as the CA's order is increased.

## C Emergence of (almost) perfect duality for higher-oder commensurate approximants

We have seen that for some special models, including the AAM, there is a $\tau$-independent duality symmetry. In these cases, the expression for the mobility edge may be obtained by using the simplest possible CA. However, this cannot be done for more generic models. Remarkably, even in such models, new duality symmetries emerge when the size of the unit cell is increased and the incommensurate limit is approached, as stated in the main text. To see that, we consider three variations of the AAM, already mentioned in the main text:

1. AA-NNN: AAM with next-nearest-neighbors;

2. $\eta$-AAM: AAM with additional staggered potential $+\eta$ in even sites and $-\eta$ in odd sites [see Fig. 23(a)];

3. 3ICS-AAM: AAM with additional different on-site energies for any three consecutive sites [see Fig. 23(b)].

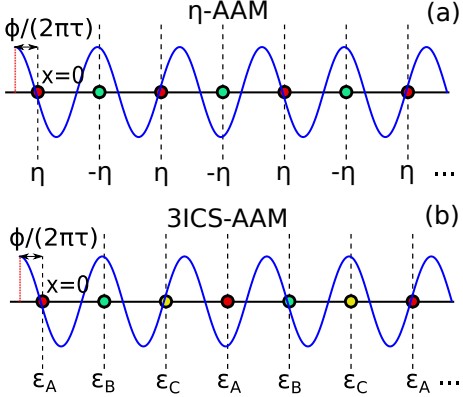

Figure 23: (a) AAM with additional staggered potential of strength $\pm\eta$ for every two consecutive sites ($\eta$-model). (b) AAM with additional on-site energies $\epsilon_A \neq \epsilon_B \neq \epsilon_C$ for every three consecutive sites (3ICS-AAM). We fix the on-site energies to $\epsilon_A = 1, \epsilon_B = -1, \epsilon_C = 0.5$, in units of $t$.

We first analyze the AA-NNN model for $\tau_c = 1$. We have

$$E = V \cos \varphi + 2 \cos \kappa + 2t_2 \cos 2\kappa \, . \qquad (50)$$

Note that for $t_2 \neq 0$ the energy dispersion is no longer invariant under a suitable interchange $\varphi \leftrightarrow k$ at $V = 2$ (and at any point).

For the $\eta$-AAM, the smallest possible unit cell corresponds to $\tau_c = 1/2$. The energy bands can be computed by solving:

$$E^2 = 2 - \frac{V^2}{2} - \eta^2 - 2\cos k - \frac{V}{2}\Big(4\eta \cos \varphi + V \cos 2\varphi\Big). \qquad (51)$$

Note that here we have identified $\varphi = \phi n_1/2$ and not $\varphi = \phi n_1$, as usual. This is due to the staggered potential, which increases the periodicity of $\phi$ by a factor of 2. For $\eta = 0$, we recover the $2\pi/n_1$-periodicity for $\phi$ and the correct definition for $\varphi$ is recovered by making $\varphi \to \varphi/2$. As implied by Eq. (51), for $\eta \neq 0$ the energy bands are never invariant under a suitable interchange $\varphi \leftrightarrow k$, for $\tau_c = 1/2$.

We start by focusing on the AA-NNN in Fig. 24. For different approximants of $\tau^{-1} = (\sqrt{5} - 1)/2$, we plot the set of points $\{(E_i, V_i)\}$ that satisfy the condition $E_i(V_i, \varphi, \kappa) = E_i(V_i, \mathcal{R}_0[\varphi, \kappa])$. In Fig. 24(a) we use $t_2 = 0.025$. In this case, the AAM's duality is only weakly broken and all the points fall into a seemingly well-defined line, to a very good approximation, even for $\tau_c = 1$. Very similar curves are obtained for different approximants, showing that the commensurate SD points quickly converge to the incommensurate SD points, even for small unit cells. This is what is expected if there are well-defined SD points for any CA: for each energy $E$, the unrotated and rotated FS fully intersect at a single coupling strength $V^*(E)$, to a very good approximation. For $t_2 = 0.2$ [Fig. 24(b)], however, the intersection points clearly fall into a region (and not a curve) for the lowest-order approximants. In this case, there is no well-defined SD curve for which the FS is invariant under $\mathcal{R}_0$, for all $\varphi, \kappa$. Interestingly, as we increase the order of the approximant the region shrinks, eventually

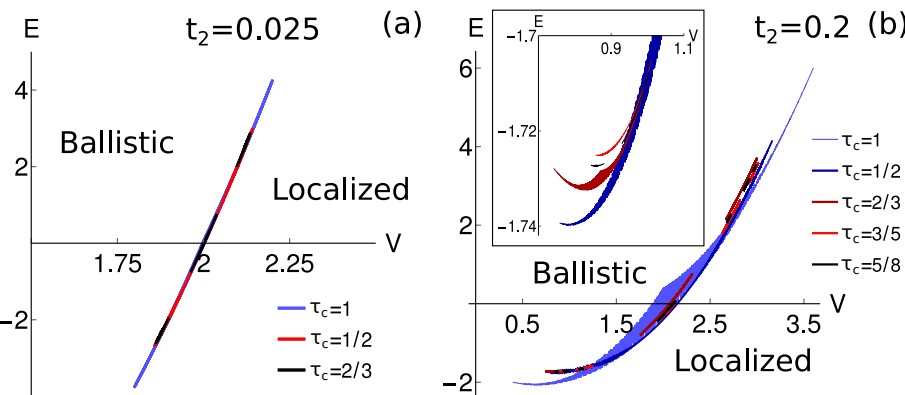

Figure 24: Set of intersection points $\{(E_i, V_i)\}$ that satisfy the condition $E_i(V_i, \varphi, \kappa) = E_i(V_i, \mathcal{R}_0[\varphi, \kappa]^T)$, for different approximants, in the AA-NNN. (a) For $t_2 = 0.025$, the duality symmetry of the AAM is only weakly broken and all these points approximately fall into a well-defined curve for any approximant. (b) For $t_2 = 0.2$ the duality symmetry of the AAM is strongly broken and the points $\{(E_i, V_i)\}$ fall into a region for approximants characterized by smaller unit cells. For such approximants, there is no true SD curve: there is no well-defined curve $E_c(V)$ for which the FS is invariant under $\mathcal{R}_0$, for all $\varphi, \kappa$. However, when the order of the approximant is increased, the region of intersection points converges into a well-defined SD curve.

looking like a curve. A perfectly well-defined SD curve is only obtained in the limit $\tau_c \to \tau$. This is also the case for $t_2 = 0.025$, in Fig. 24(a), although in this case the curve seems essentially perfect within numerical accuracy, even for very small unit cells. This is because in practice, CA of high-enough order already provide an essentially perfect description of the incommensurate thermodynamic limit SD curve. In Fig. 24(b), for instance, the intersection points already fall into an almost perfect curve for $\tau_c = 5/8$.

For the $\eta$-AAM, the main results are explained in Fig. 25. For $\tau_c = 1/2$, we can see that the energy bands are almost (but not exactly) $\varphi$-periodic with period $\Delta\varphi = \pi$ [Fig. 25(a)]. This is expected because we have used a small $\eta = 0.1$ and therefore the model is similar to the AAM (where $\varphi \to 2\varphi$). In this case it is obvious that no SD symmetry can be found [Fig. 25(c)]. For higher-order approximants, however, the almost $\pi$-periodicity is completely lost [Fig. 25(b)] and a SD symmetry with the correct $2\pi$-periodicity emerges [Fig. 25(d)].

Note that for small $\eta$ the mobility edges seem to vary in a chaotic manner, as illustrated in Fig. 25(e). The reason is that the $\eta$-perturbation defines a new periodicity for $\phi$, different from the one in the parent model. The consequence is that the duality symmetry also becomes completely different from the one in the parent model. With the correct definition of $\varphi = \phi n_1/2$, only the fundamental harmonic [$\cos(\varphi)$] survives for a CA of high-enough order. However, for low-order CA and small $\eta$, the system is still very similar to the AAM: the universal dualities will only appear for high-order CA and may give rise to significantly different SD curves (mobility edges), even for bands that are close in energy. If we instead consider a large $\eta$, the mobility edge is almost entirely well captured for lower-order CA. In this case, the $\phi$-periodicity of the AAM is completely broken even for the lower-order CA: the almost perfect SD symmetry with the correct periodicity emerges even for these. An example is in Fig. 26, for $\eta = 1.5$.

We finish this section by analyzing the 3ICS-AAM. Differently from the $\eta$-model, this model does not host any obvious $\phi^*$ around which the system's properties are symmetric. For the $\eta$-model, $\phi^* = 0$ is obviously one of such points. In that case, the FS of CA in the $(\varphi, \kappa)$ plane can be described by simple cossines, with $\varphi_0 = 0$ (FS is even around $\varphi_0 = 0$, due to the symmetry

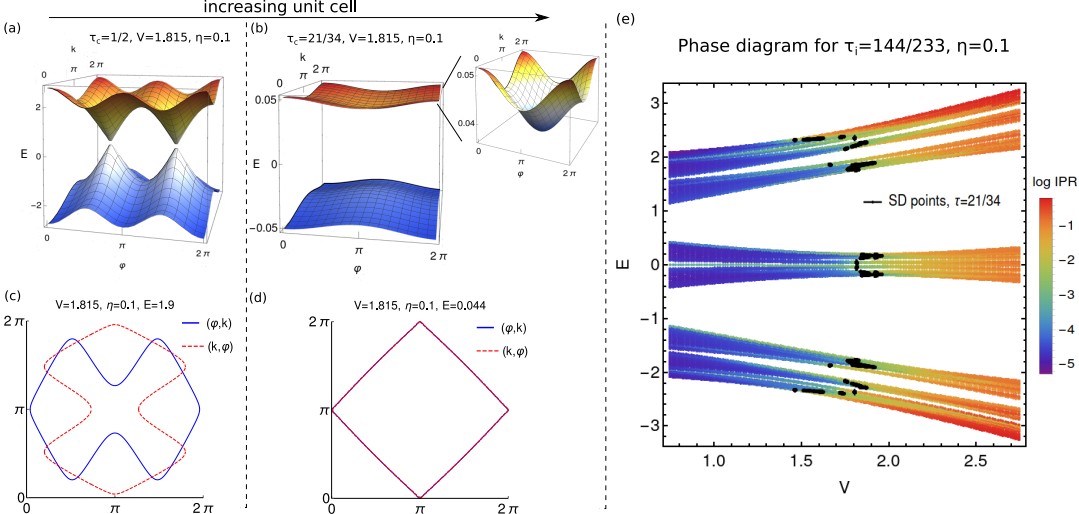

Figure 25: Results for the $\eta$-model, with $\eta = 0.1$. (a,b) $E(\varphi, \kappa)$ energy bands for $\tau_c = 1/2$ and $\tau_c = 21/34$. For the latter, the two bands closer to $E = 0$ were plotted. The inset shows a close-up of one of the bands. (c,d) Examples of constant energy cuts of the $E(\varphi, \kappa)$ bands. For $\tau_c = 1/2$, there is no cut for which the SD symmetry is observed. For $\tau_c = 21/34$ such cuts exist for most of the bands. (e) Spectrum and IPR for $\tau_i = 144/233$ along with predicted SD points for $\tau_c = 21/34$.

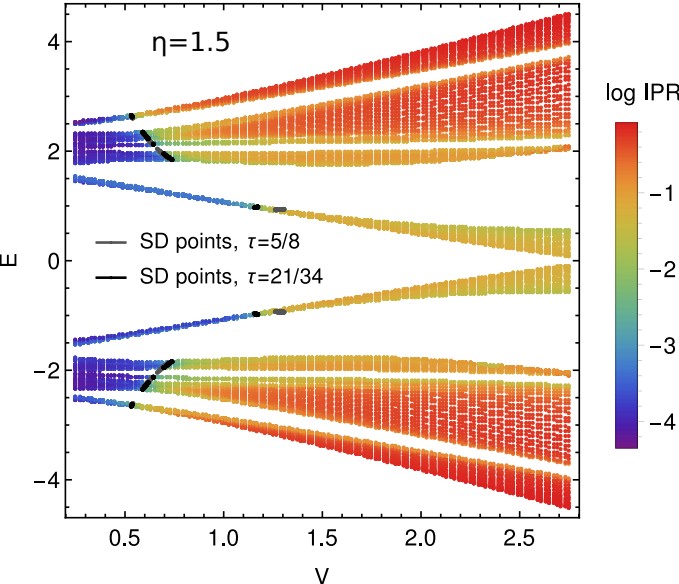

Figure 26: Spectrum and IPR for $\tau_i = 144/233$ along with predicted SD points for $\tau_c = 5/8$ (gray) and $\tau_c = 21/34$ (black).

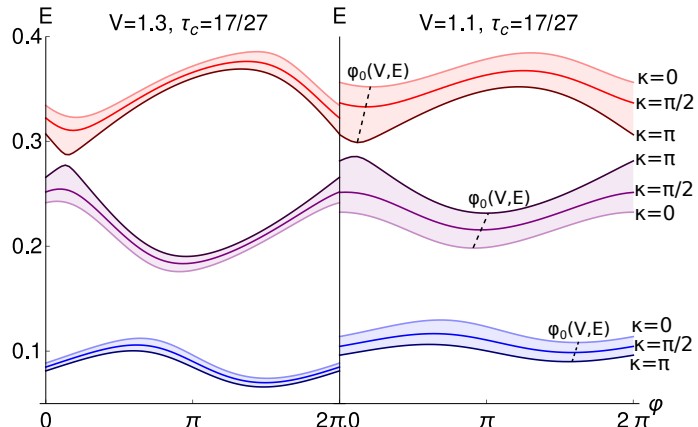

Figure 27: Energy bands of the CA defined by $\tau_c = 17/27$ for the 3ICS-AAM , as a function of $\varphi$ and for different $V$. For each band, we plot contours corresponding to $\kappa = 0, \pi/2, \pi$.

around $\phi_0 = 0$). In the new model, shifting the potential to negative $\phi$ (towards $\epsilon_C$) is not equivalent to shifting it to positive $\phi$ (towards $\epsilon_B$). This results in an important qualitative change with respect to previous models that we studied: $(\varphi_0, \kappa_0)$ acquires a dependence on energy and on the Hamiltonian's parameters.

We start by noticing that the energy bands of the 3ICS-AAM are periodic with respect to $\phi$-translations, with period $3 \times 2\pi/n_1$. Notice the factor of 3 that comes from the existence of three consecutive inequivalent sites. This motivates the definition of $\varphi$ as $\varphi = n_1 \phi/3$. For the following results, we consider $\epsilon_A = 1, \epsilon_B = -1, \epsilon_C = 0.5$.

In Fig. 27 we show some energy bands as a function of $\varphi$. $\varphi_0$ can be seen as the phase $\varphi$ corresponding to the minimum of a given energy band, for a fixed $\kappa$. As previously stated, we see that $\varphi_0$ depends on the Hamiltonian's parameters and on energy. For previous models, we always obtained $\varphi_0 = 0$ or $\varphi_0 = \pi$. This was a consequence of the models' symmetry under shifts around these points. In the present case, there are no obvious points for which the system is symmetric under shifts.

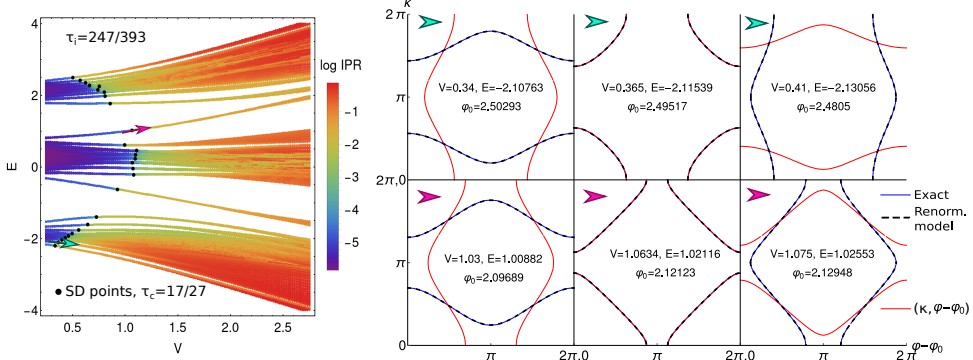

Figure 28: Left: IPR for the 3ICS-AAM, for a QPS with $\tau_i = 247/393$ (system with 393 sites), together with SD points computed for the CA with $\tau_c = 17/27$. Right: Examples of FS around the critical point. The blue and red lines correspond to the exact FS, respectively with $\varphi - \varphi_0$ and $\kappa$ in the x-axis. The dashed black lines correspond to the results of the renormalized model in Eq. 39. Note that in the middle panel, the red and blue lines are superimposed because these FS were computed at SD points.

Remarkably, we see that the FS again become perfectly sinusoidal around the critical point, but now with a shift $\varphi_0$ that varies smoothly with the model's parameters. This can be seen in Fig. 28. There, we start by showing the IPR for a QPS with $\tau_i = 247/393$, that shows a clear transition from an extended to a localized phase. In black, we plot SD points for a CA with $\tau_c = 17/27$. These were computed by shifting the FS by $-\varphi_0(V, E)$ in the $\varphi$ direction and then computing the renormalized parameters as described in Sec. B.2. Some examples of FS are shown inthe right panel of Fig. 28, where we can see the perfect description of the FS by the model in Eq. 39 and the existence of SD points that match the localization-delocalization transition.

Finally, we emphasize that in general the simple sinusoidal description of the FS is only valid around the critical point. The current model is an example of that: if we look at the energy bands away from criticality, in particular for large-$V$, we see a lot of non-trivial $\varphi$-driven band-crossings [see Fig. 29(c)]. In such cases, the FS is more complicated (even though being non-dispersive in $\kappa$). Nonetheless, sufficiently close to the critical point, no band crossings should occur: at criticality, band-gaps are opened at any CA's order [7].

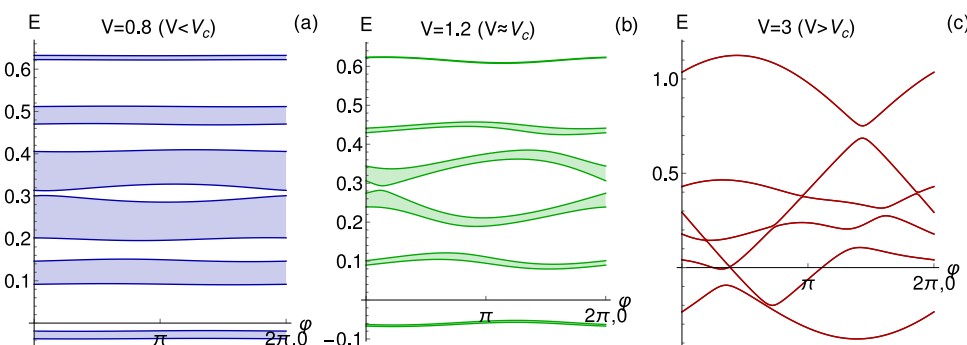

Figure 29: Examples of energy bands of the 3ICS-AAM. These bands are integrated over $\kappa$ and plotted for variable $\varphi$ and different $V$ below (a), close-to (b) and above (c) criticality.

---

[7]This follows from the infinite correlation length at the critical point. Increasing the order of the approximant

# D  Derivation of Eq. 35

In the main text we claimed that the Hamiltonian in Ref. [34] is self-dual under the following duality transformation,

$$f_k = \sum_p e^{-2\pi i \tau k p} \chi_p(\beta_0) u_p \,, \tag{52}$$

where $\chi_p(\beta) = \frac{\sinh\beta}{\cosh\beta - \cos(2\pi\tau p)}$, $\cosh\beta = \alpha^{-1}$ and $\beta_0$ is defined as $2t\cosh\beta_0 = E + 2\lambda\cosh\beta$. Here we prove that statement. The Schrodinger equation for this model can be written as (Eq. (7) of Ref. [34]):

$$t(u_{p-1} + u_{p+1}) + g\chi_p(\beta)u_p = (E + 2\lambda\cosh\beta)u_p \,. \tag{53}$$

We know from Ref. [34] that the self-duality condition implies that $\beta = \beta_0$. The derivation is made below term by term:

**Term $t(u_{p-1} + u_{p+1})$.**

Starting with $tu_{p-1}$, we multiply by $e^{-2\pi i \tau k p}$ and sum over $p$,

$$
\begin{aligned}
tu_{p-1} &\to \sum_p tu_{p-1} e^{-2\pi i \tau k p} \chi_{p-1}^{-1}(\beta_0) \chi_{p-1}(\beta_0) \\
&= \frac{1}{\sinh\beta_0} \sum_p tu_{p-1} [\cosh\beta_0 - \cos[2\pi\tau(p-1)]] e^{-2\pi i \tau k(p-1)} \chi_{p-1}(\beta_0) e^{2\pi i \tau k} \\
&= \frac{1}{\sinh\beta_0} \sum_p tu_p [\cosh\beta_0 - \cos(2\pi\tau p)] e^{-2\pi i \tau k p} \chi_p(\beta_0) e^{2\pi i \tau k} \,.
\end{aligned}
\tag{54}
$$

In the expression obtained above, the first term becomes $\frac{t\cosh\beta_0}{\sinh\beta_0} f_k e^{2\pi i \tau k}$, while the second one is $-\frac{t}{2\sinh\beta_0}(f_{k+1} + f_{k-1}) e^{2\pi i \tau k}$. In a similar way, for $tu_{p+1}$,

$$
\begin{aligned}
tu_{p+1} &\to \sum_p tu_{p+1} e^{-2\pi i \tau k p} \chi_{p+1}^{-1}(\beta_0) \chi_{p+1}(\beta_0) \\
&= \frac{t\cosh\beta_0}{\sinh\beta_0} f_k e^{-2\pi i \tau k} - \frac{t(f_{k+1} + f_{k-1})}{2\sinh\beta_0} e^{-2\pi i \tau k} \,.
\end{aligned}
\tag{55}
$$

Combining the two terms,

$$t(u_{p-1} + u_{p+1}) \to -\frac{t\cos(2\pi\tau k)}{\sinh\beta_0}(f_{k+1} + f_{k-1}) + \frac{2t\cosh\beta_0}{\sinh\beta_0}\cos(2\pi\tau k)f_k \,. \tag{56}$$

**Term $g\chi_p(\beta)u_p$.**

Multiplying by $e^{-2\pi i \tau k p}$ and summing over $p$,

$$g\chi_p(\beta)u_p \to g\sum_p \chi_p(\beta)\chi_p^{-1}(\beta_0)u_p e^{-2\pi i \tau k p}\chi_p(\beta_0) \,. \tag{57}$$

We can simplify the result here by using *a priori* the self-duality condition $\beta = \beta_0$. In that case, this term simply becomes $gf_k$.

---

can reflect in changes that are only observed at very large length scales. But if the correlation length is infinite, these changes will always be relevant and gaps will be opened at any order.

**Term** $(E + 2\lambda \cosh \beta) u_p$.

Again, we multiply by $e^{-2\pi i \tau k p}$ and sum over $p$,

$$(E + 2\lambda \cosh \beta) u_p \rightarrow \sum_p (E + 2\lambda \cosh \beta) \chi_p^{-1}(\beta_0) \chi_p(\beta_0) \times u_p e^{-2\pi i \tau k p} \,. \tag{58}$$

Recalling that $\chi_p^{-1}(\beta_0) = \frac{\cosh \beta_0 - \cos(2\pi \tau p)}{\sinh \beta_0}$, the first term becomes

$$\frac{\cosh \beta_0}{\sinh \beta_0} \sum_p (E + 2\lambda \cosh \beta) \chi_p(\beta_0) u_p e^{-2\pi i \tau k p} = \frac{\cosh \beta_0 (E + 2\lambda \cosh \beta)}{\sinh \beta_0} f_k \,, \tag{59}$$

and the second term,

$$-\frac{1}{\sinh \beta_0} \sum_p (E + 2\lambda \cosh \beta) \cos(2\pi \tau p) u_p e^{-2\pi i \tau k p} = -\frac{E + 2\lambda \cosh \beta}{2 \sinh \beta_0} (f_{k+1} + f_{k-1}) \,. \tag{60}$$

**Combine everything**

Combining everything, we have

$$\begin{aligned} \Big[ -\frac{t \cos(2\pi \tau k)}{\sinh \beta_0} + \frac{E + 2\lambda \cosh \beta}{2 \sinh \beta_0} \Big] (f_{k+1} + f_{k-1}) + g f_k \\ = \frac{\cosh \beta_0}{\sinh \beta_0} [E + 2\lambda \cosh \beta - 2t \cos(2\pi \tau k)] f_k \,. \end{aligned} \tag{61}$$

We now verify that for the duality condition $\beta = \beta_0$, the model above is self-dual of the model in Eq. (53). This condition implies that $E + 2\lambda \cosh \beta = 2t \cosh \beta$. The term with square brackets on the left hand side of Eq. (53) becomes

$$\frac{t}{\sinh \beta} \Big[ -\cos(2\pi \tau k) + \cosh \beta \Big] (f_{k+1} + f_{k-1}) = t \chi_k^{-1}(\beta) (f_{k+1} + f_{k-1}) \,, \tag{62}$$

while the right hand side can be written as

$$\frac{\cosh \beta}{\sinh \beta} [E + 2\lambda \cosh \beta - 2t \cos(2\pi \tau k)] f_k = 2t \cosh \beta \chi_k^{-1}(\beta) f_k \,. \tag{63}$$

Combining everything again:

$$\begin{aligned} t \chi_k^{-1}(\beta)(f_{k+1} + f_{k-1}) + g f_k &= 2t \cosh \beta \chi_k^{-1}(\beta) f_k \\ \Leftrightarrow t(f_{k+1} + f_{k-1}) + g \chi_k(\beta) f_k &= (E + 2\lambda \cosh \beta) f_k \,, \end{aligned} \tag{64}$$

which is clearly dual of Eq. (53).

# E  Additional remarks on wave function duality

We have seen in the main text that an existing duality symmetry maps the real-space wave function $\psi^r(\phi, k)$ and its dual $\psi^d(\mathcal{R}_0[\phi, k])$, where $\mathcal{R}_0$ is a $\pi/2$-rotation in the $(\phi, k)$ plane around point $(\phi_0, k_0)$. In the text, we mostly dealt with examples for which $\phi_0 = 0$ or $\phi_0 = \pi$. In this section we show that this 'center of rotation' can be less trivial.

In general it may be challenging to find the correct 'center of rotation' $(\phi_0, k_0)$. The knowledge of such point is essential to correctly define the duality transformation for the wave function. A possible quantity that can hint this point is the overlap between $\tilde{\psi}^r(\phi, k)$ and $\tilde{\psi}^d(\phi, k)$

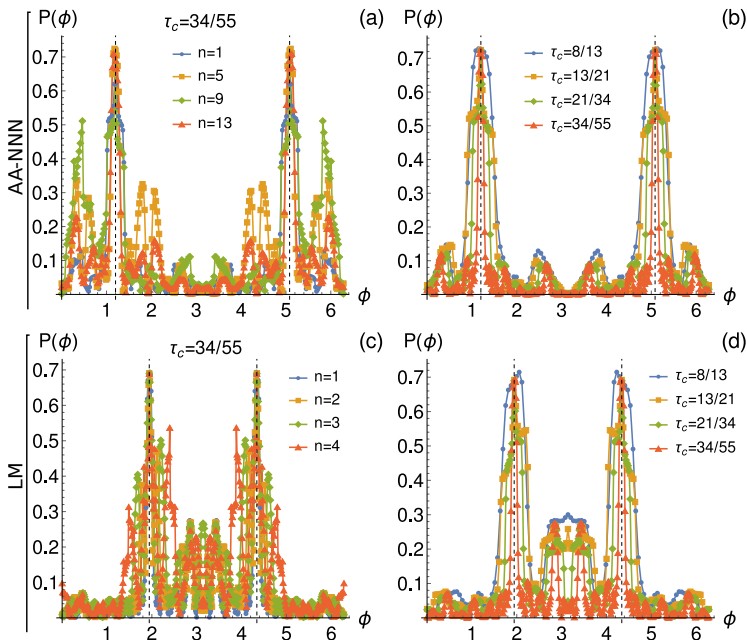

Figure 30: Overlap $P(\phi) = |\langle \tilde{\psi}^{\mathrm{r}}(\phi, k=0) | \tilde{\psi}^{\mathrm{d}}(\phi, k=0) \rangle|$, computed at SD points. (a,b) Results for the AA-NNN, with $t_2 = 0.5$. In (a) $\tau_c = 34/55$ is fixed and $P(\phi)$ is computed for some SD points within the interpolated SD curve with negative concavity in Fig. 2(d). In (b), we vary $\tau_c$ and compute $P(\phi)$ for the lowest-energy SD point. The dashed lines correspond to $\phi = \pi(n_1 - n_2)/n_1 \lor \phi = 2\pi - \pi(n_1 - n_2)/n_1$. (c,d) Results for the LM, with $V = 3, \Lambda = 2.5$. In (c) we fix $\tau_c = 34/55$ and use SD points within the lowest-energy interpolated SD curve with negative concavity in Fig. 2(c). In (d), we vary $\tau_c$ and compute $P(\phi)$ for the lowest-energy SD point. The dashed lines correspond to $\phi = \pi n_2/n_1 \lor \phi = 2\pi - \pi n_2/n_1$.

at dual points, for variable $(\phi, k)$. A large overlap may be found at point $(\phi_0, k_0)$ which is, by definition, invariant upon application of the duality transformation. In fact, for the examples that we tested, the real-space wave function and its Aubry-André dual were always similar, which was reflected in a relatively large overlap. With this in mind, we define the quantity,

$$P(\phi) = |\langle \tilde{\psi}^{\mathrm{r}}(\phi, k=0) | \tilde{\psi}^{\mathrm{d}}(\phi, k=0) \rangle|, \tag{65}$$

as the overlap between $\tilde{\psi}^{\mathrm{r}}(\phi, k)$ and $\tilde{\psi}^{\mathrm{d}}(\phi, k)$, at SD points, for $k = 0$ (we have $k_0 = 0$ in the examples that follow and only need to search for $\phi_0$). For fixed $k_0 = 0$, we define $\phi_0$ as the phase that maximizes $P(\phi)$, checking that it remains stable as the CA's order is increased. In the cases shown in the main text (Figs. 11-13), $\phi_0 = 0$ or $\phi_0 = \pi$. In Fig. 30, we show examples for the AA-NNN and LM, for which we identified, respectively $\phi_0 = \pi(n_1 - n_2)/n_1 \lor \phi_0 = 2\pi - \pi(n_1 - n_2)/n_1$ and $\phi_0 = \pi n_2/n_1 \lor \phi_0 = 2\pi - \pi n_2/n_1$. For the AA-NNN these phases are for critical states within the SD curve with negative concavity in Fig. 2(d). For the LM, we presented the center of rotation for the lowest-energy interpolated SD curve with negative concavity, see Fig. 2(c). For the mentioned $\phi_0$ phases, $P(\phi_0)$ is a stable maximum in the sense that it remains a maximum when the CA's order is increased.

We finish by alerting that for some cases, the wave function and its Aubry-André dual may be distinct - the generalized duality may be very different from the Aubry-André duality. In such cases, we have no systematic way of defining the duality transformation for the wave function. The mobility edge, on the other hand, may still be predicted. For the latter, we work with variables $\varphi$ and $\kappa$ which allow us to find SD points.

# F  Possible generalizations: example of Maryland model

In the models that we have seen the FS in the $(\varphi, \kappa)$ space can be well described by Eq. 19 at each energy. However, the effective model describing FS can be different, even though the treatment in terms of renormalized couplings that measure the dependence on $\varphi$ and $\kappa$ still holds. Example that we will treat in detail in [54] are models that also contain a coupling $C_R \cos(\varphi) \cos(\kappa)$. These, as we will show, are responsible for the existence of phases with critical eigenstates that exist over a region of parameters (and not only at fine-tuned points such as at the critical point between a delocalized and a localized phase). Here we show yet another example in which the effective model is different than the one in Eq. 19: the Maryland model [59]. This model has an unbounded quasiperiodic potential with Hamiltonian is given by

$$H = V \sum_n \tan[(2\pi\tau n + \phi)/2] c_n^\dagger c_n + \sum_n \left( c_n^\dagger c_{n+1} + \text{h.c.} \right). \tag{66}$$

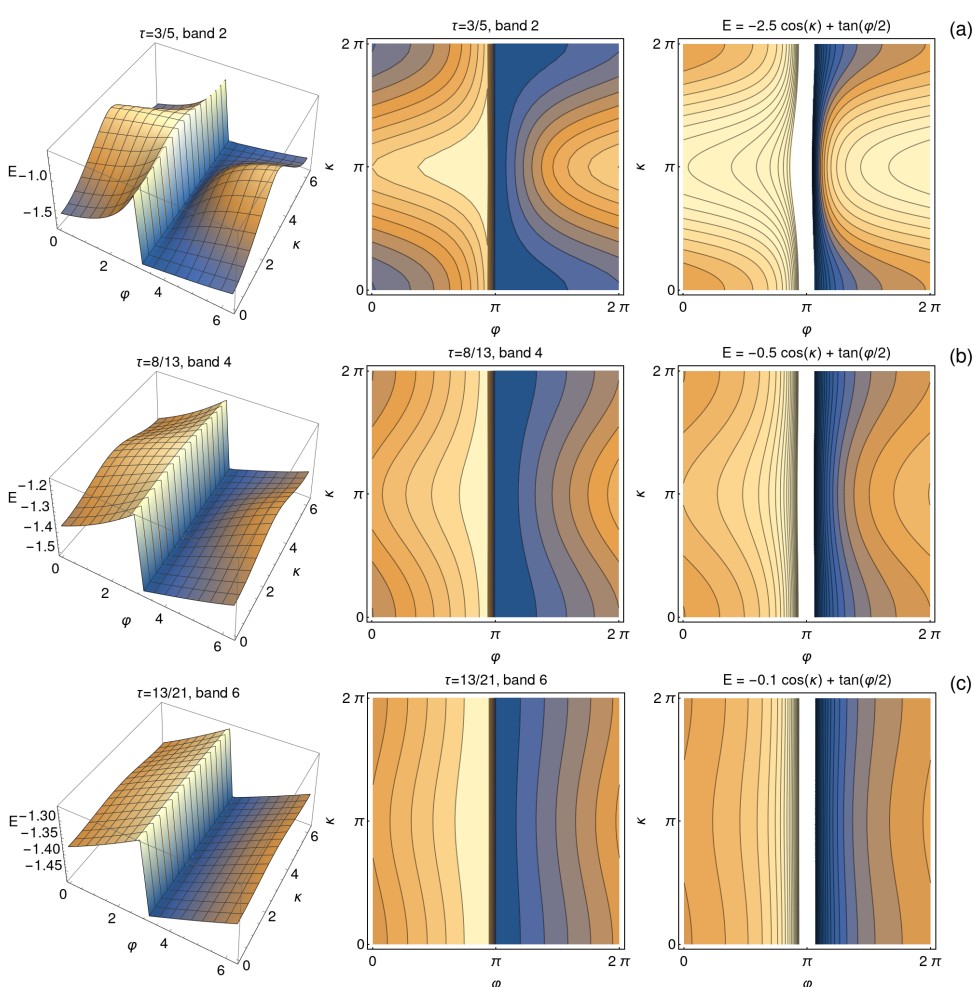

Figure 31: Energy bands and FS as a function of $\varphi = n_1 \phi$ and $\kappa$ for selected energy bands of CA of the model in Eq. 66. (a) $\tau_c = 3/5$, 2nd band (ordered from lowest to higher energy); (b) $\tau_c = 8/13$, 4th band; (c) $\tau_c = 13/21$, 6th band. The rightmost plot of each sub-figure shows how the effective model in Eq. 67 can describe the FS obtained numerically for the different CA.

.

In this case, the FS for different CA can be captured through

$$E = V_R(V,E)\tan(\varphi) + 2t_R(V,E)\cos(\kappa) + E_R(V,E), \tag{67}$$

as exemplified in Fig. 31. Indeed, in the Maryland model, the system becomes localized for any finite $V$. Therefore, for a large enough unit cell the coupling $t_R$ becomes irrelevant with respect to the coupling $V_R$. We show in Fig. 31 that this is the case using an example for $V = 0.2$.

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
