# Peer review of "Hidden dualities in 1D quasiperiodic lattice models"

_SciPost Physics, doi:SciPost Phys. 13, 046 (2022)_

## Round 3 · Referee Report · Anonymous (Referee 1) · 2022-4-23

Report

This is a report on the manuscript “Hidden dualities in 1D quasiperiodic lattice models” by Miguel Goncalves and co-worders.
In this paper, the authors study the localization transition in various extended Aubry-Andre and an incommensurate ladder model. These models have been chosen to contain nontrivial mobility edges, and the main finding of this work is that the localization transitions in all studied cases and at all energies are controlled by dualities. These dualities, which the authors term hidden dualities, are introduced as generalizations of the well-known Aubry duality of the Aubry-Andre model.
Their findings are substantiated by analytical arguments as well as strong numerical data for the studied models, and the authors conjecture that these hidden dualities might be a generic feature of quasi-periodic models.
Quasiperiodic models are currently being studied extensively in various contexts ranging from mathematical physics to the (also experimental) study of many-body localization. As these findings represent a significant step forward in categorizing and understanding the underlying transitions in the non-interacting case, they will be read with interest within these communities.
The paper is written clearly, and I am happy to recommend publication following some minor edits.

Requested changes

In particular, on page 6, right column, the authors state that “Figure 6(b) also suggests that FS for E >Ec(V ) can be mapped into FS for E <Ec(V ) upon this rotation, with Ec (V ) the energy at the SD point.” At this point in the text, this is not obvious to me and maybe should be expanded on.
Additional minor points:
• Page 2: “reminder of this section” -> section II
• Fig 1, caption: “Black shaded” -> dark grey shaded. What does the light grey shading indicate? (it is not yet explained in the caption). Figures a,b use labels V and V’; but c,d and the caption use V’ and V’’. I assume that figures c,e correspond to tau_c=1 and d,f to tau_c=2/3 – this should be clarified in the caption.
• In Fig 2, the parameter xi is not introduced yet. It is not obvious that the label E and V in figures (e) correspond to the x and y axes. It is also initially confusing that both plots in (e) [and (f)] correspond to the system shown in (b) and (d); one would expect that the left pictures correspond to (a) and (c).
• Page 3, right column, top line should refer to fig 2 and not fig 1.
• Fig 3, caption: latter -> ladder
• Fig 4: what is the purpose of the grey and magenta shaded regions? Delete or explain in caption
• Page 7, left column: “dependent,on” replace “,” by “space”
• Page 7, right column: “.except”: superfluous “.”

  • validity: high
  • significance: good
  • originality: top
  • clarity: high
  • formatting: excellent
  • grammar: excellent

Author:  Miguel Gonçalves  on 2022-06-07  [id 2566]

(in reply to Report 1 on 2022-04-23)

We thank the Referee for the very positive feedback on our manuscript. We are glad to know the Referee finds our work "a significant step forward in categorizing and understanding the underlying transitions (in quasiperiodic models) in the non-interacting case". We also thank the Referee for spotting typos that could lead to confusion for the reader and providing useful suggestions.
We clarified all the points raised by the Referee, as detailed below.

""In particular, on page 6, right column, the authors state that “Figure 6(b) also suggests that FS for E >Ec(V ) can be mapped into FS for E <Ec(V ) upon this rotation, with Ec (V ) the energy at the SD point.” At this point in the text, this is not obvious to me and maybe should be expanded on.""

R: We clarified this at the second paragraph of sub-section "Close to critical point", in section V.

""Fig 1, caption: “Black shaded” -> dark grey shaded. What does the light grey shading indicate? (it is not yet explained in the caption). Figures a,b use labels V and V’; but c,d and the caption use V’ and V’’. I assume that figures c,e correspond to tau_c=1 and d,f to tau_c=2/3 – this should be clarified in the caption.""

R: We corrected the typos and clarified figures c,e in the caption. We also explained the light grey shading in the figure's caption.

""In Fig 2, the parameter xi is not introduced yet. It is not obvious that the label E and V in figures (e) correspond to the x and y axes. It is also initially confusing that both plots in (e) [and (f)] correspond to the system shown in (b) and (d); one would expect that the left pictures correspond to (a) and (c).""

R: We implemented all the suggestions. To make Figs.(e,f) less confusing, we added a dashed box around them that connects with Fig.(d).

""Fig 4: what is the purpose of the grey and magenta shaded regions? Delete or explain in caption""

R: We added an explanation to the caption.

---

## Round 3 · Referee Report · Anonymous (Referee 2) · 2022-5-6

Strengths

This paper is able to generalize the basic intuition for particular quasiperiodic models with localization transitions to a wide class of systems. It is well written, clearly presented, and I think an important advance.

Weaknesses

I don't see any weaknesses with the paper.

Report

In this manuscript the authors have discovered a way to generalize the notion of a duality between extended and localized phases in non-interacting one-dimensional quasiperiodic systems. The self-dual nature of the Aubry-Andre model at its critical point as well as duality mappings between extended and localized phases are well known and there have been extensions to this model that make this duality energy dependent inducing a mobility edge. However, in general identifying such a duality around quasiperiodic induced localization transition has remained elusive though such ideas have been discussed informally in the community. I was very excited to work through this manuscript as the authors have successfully completed this idea, put it onto solid mathematical ground, and demonstrated its utility across several models that possess mobility edges and do not have an exact analytic duality. The procedure to identify a hidden duality is clearly presented, and when compared with numerical results shows great agreement even on small commensurate approximants. I am happy to recommend this paper for publication in SciPost as I think the results of this paper are significant and I expect that it will have a strong impact on the field of quasiperiodic systems.

I think some aspects of the presentation can be streamlined for the reader a bit as I detail below as well as a present a few questions that the authors should discuss in a revised manuscript.

Requested changes

1) As there are several models studied it would be helpful for the reader to always specify the model and parameters used, while also reminding the reader about the physical content of each (e.g. \xi is the hopping decay length, V is the strength of the potential, x is…. etc). Along these lines it would also be helpful to reference equations for certain quantities as they are discussed in each Fig caption, such as \chi(\beta0, x), \chi_c, ect. While the authors do a reasonable job at this in the present draft I think it can be improved further to improve the overall readability.
a) Also, for the LM model V_p is used when defining t_{\perp} but it is then later called V.

2) When the authors specify that they can estimate the localization transition with good accuracy using relatively small rational approximants, I think its important to show the quality of this convergence quantitatively, in other words, how accurate is the estimate as the system size increases? I would therefore suggest that they present a more detailed comparison between the computed self-dual line for a rational approximant and their estimate of the transition from finite size scaling. Such a comparison would be quite useful to show the accuracy of their approach beyond comparing the self-dual lines to color plots of the inverse participation ratio.

3) Does this approach work if the quasiperiodic potential is unbounded? For example, the Maryland mode, or the GAAM in the limit |alpha|>1 (to follow up on a recent study for example)?

4) How well does this approach work when considering a Fibbonacci potential? It would be interesting to explore thisusing the potential in the model presented in Ref. Phys. Rev. Research 3, 033257 (2021).
.

5) Do the authors envision any analytic approach beyond their numerical “recipe”? It would be interesting to discuss this within a systematic perturbative/self consistent framework such as in Refs. Phys. Rev. B 104, 064201 (2021). Such a perturbative approach could be applied in either phase by working in either real or momentum space basis.

6) Have the authors thought through if this approach can also be naturally applied quasiperiodic systems that have a topological band structure (such as Su-Schrieffer-Hegger chains, 2D/3D topological insulators and semimetals)? I am wondering if the presence of edge states has any effect on their procedure as they could potentially alter the Fermi surface analysis.

  • validity: top
  • significance: top
  • originality: top
  • clarity: top
  • formatting: perfect
  • grammar: perfect

Author:  Miguel Gonçalves  on 2022-06-07  [id 2567]

(in reply to Report 2 on 2022-05-06)

We thank the Referee for the very positive feedback on our manuscript. We are pleased to know that the Referee finds that the "results of this paper are significant and" expects "that it will have a strong impact on the field of quasiperiodic systems". Below we address in detail all the questions/suggestions made by the Referee.

""1) As there are several models studied it would be helpful for the reader to always specify the model and parameters used, while also reminding the reader about the physical content of each (e.g. \xi is the hopping decay length, V is the strength of the potential, x is…. etc). Along these lines it would also be helpful to reference equations for certain quantities as they are discussed in each Fig caption, such as \chi(\beta0, x), \chi_c, ect. While the authors do a reasonable job at this in the present draft I think it can be improved further to improve the overall readability.
a) Also, for the LM model V_p is used when defining t_{\perp} but it is then later called V.""

R: Throughout the manuscript, we added to each figure caption reminders to the reader of the model and parameters being studied, and added hyperlinks to the relevant equations where these are defined.

""2) When the authors specify that they can estimate the localization transition with good accuracy using relatively small rational approximants, I think its important to show the quality of this convergence quantitatively, in other words, how accurate is the estimate as the system size increases? I would therefore suggest that they present a more detailed comparison between the computed self-dual line for a rational approximant and their estimate of the transition from finite size scaling. Such a comparison would be quite useful to show the accuracy of their approach beyond comparing the self-dual lines to color plots of the inverse participation ratio.""

R: We thank the referee for the very useful suggestion. We added an inset to Fig.2(b) an example with results of the prediction of self-dual points through commensurate approximants having different unit cell sizes, in comparison with the estimation of the critical point through the IPR for a large system size. The aim is to clearly show that predictions of self-dual points for small unit cells can already provide very accurate predictions of the critical point.

""3) Does this approach work if the quasiperiodic potential is unbounded? For example, the Maryland mode, or the GAAM in the limit |alpha|>1 (to follow up on a recent study for example)?""

R: For the GAAM when |alpha|>1, the eigenstates can be critical over an extended region of parameters. The scope of our work was to characterize localization-delocalization transitions, for which critical states occur only at the critical point separating the two phases. Nonetheless, we are now working on generalizing our approach to capture critical phases also and preliminary results indicate that such generalization is indeed possible. Our findings on this matter will be made public as soon as possible. To clarify this issue within the scope of the present manuscript, we added some comments to the Discussion section.
For the Maryland model we found that interestingly the effective model that describes the Fermi surfaces in terms of \varphi and \kappa for different approximants involves a tangent of \varphi and not a cossine. In this case, there is no duality-driven transition since the eigenstates become localized for any intensity of the quasiperiodic potential. This is well captured within our approach - when the size of the unit cell is increased, the renormalized coupling that multiplies the \kappa-dependent term becomes irrelevant with respect to the coupling that multiplies the \varphi-dependent term. We added a small section to the Appendices regarding this model since it is an interesting case in which even though there are no duality-driven transitions, the localization properties can still be captured with the ideas introduced in our work. We also added some comments in the Discussion.

""4) How well does this approach work when considering a Fibbonacci potential? It would be interesting to explore thisusing the potential in the model presented in Ref. Phys. Rev. Research 3, 033257 (2021).""

R: This is also an interesting line to further explore. We thank the referee for the nice suggestion of using the interpolating potential between the Aubry-André and Fibbonacci model. We checked that, for instance, the localization-delocalization transition in Fig.2(a) of Phys. Rev. Research 3, 033257 (2021) is well-captured by our hidden dualities (see "response_aux.pdf" in attachment).
In the Fibbonacci model, the eigenstates are critical, independently of the strength of the potential. This is beyond the scope of our manuscript, but it will be interesting to see whether the generalization that we are working on to capture critical phases (mentioned in the previous question) can capture the Fibbonacci's model critical phase.

""5) Do the authors envision any analytic approach beyond their numerical “recipe”? It would be interesting to discuss this within a systematic perturbative/self consistent framework such as in Refs. Phys. Rev. B 104, 064201 (2021). Such a perturbative approach could be applied in either phase by working in either real or momentum space basis.""

R: Yes and we are currently working on that. We have preliminary results indicating that in some models it is possible to analytically calculate renormalized couplings that measure how the energy dependence on phase twists and phase shifts evolves with the size of the unit cell. In such cases, the phase diagram can therefore be obtained analytically. In more generic models the analytical calculation of the renormalized couplings can be made only for small unit cells. However in some cases, perturbative analytical approximations of the phase diagram can also be made through the knowledge of the latter. We are now working on understanding exactly in which models exact or approximate analytical calculations of the phase diagram can be made and will latter make all our new findings public.

""6) Have the authors thought through if this approach can also be naturally applied quasiperiodic systems that have a topological band structure (such as Su-Schrieffer-Hegger chains, 2D/3D topological insulators and semimetals)? I am wondering if the presence of edge states has any effect on their procedure as they could potentially alter the Fermi surface analysis.""

R: We believe that our approach can also be applied to 1D systems with non-trivial topology. We have indeed tested some examples of SSH models with quasiperiodic potentials for which localization-delocalization transitions are well captured by dualities.
In fact the existence of edge states should not influence our treatment since our generalized dualities are defined for systems with periodic (twisted) boundary conditions and are therefore a bulk property.
An interesting additional question is what is the connection between the generalized dualities and the non-trivial topological properties that are intrinsic to quasiperiodic systems. These can be characterized by a Chern number defined in the space of phase twists and an additional virtual dimension corresponding precisely to the phase shifts that we use in our description. We plan to address this question in the future.

We added some comments to the Discussion section regarding the Referee's questions.

Attachment:

response_aux.pdf

---

## Round 4 · Referee Report · Anonymous (Referee 3) · 2022-6-9

Report

The authors have responded to each question in turn. I am excited to see some of these answers in more detail in their future. work.

---

## Round 4 · Referee Report · Anonymous (Referee 4) · 2022-7-2

Report

The revised version addresses all minor points I raised previously; I have no additional comments and would be happy to see the paper published.

---

## Round 4 · Author Response

Dear Editor,

Thank you very much for your time and consideration of our manuscript. We are grateful for the very positive Referee reports on our work and for the very useful suggestions and comments that were made by you and the Referees.
In our revised manuscript, we have corrected all the typos that you and the Referees noticed. We also tried to improve accessibility and readability of the manuscript. In particular, we added to each figure caption reminders of the model and parameters being studied in that figure, and added hyperlinks to the relevant equations where these were defined. We also specified the Appendix sections that we are referring to whenever they are mentioned in the main sections for a faster navigation throughout the manuscript.
On more specific details, regarding section II, we have both the "Aubry-André duality" and "General dualities" sub-sections.
In our responses to the Referee reports, we addressed all the points raised by the Referees and indicated the changes that we made in the manuscript regarding these questions, whenever applicable.

Yours sincerely,
Miguel Gonçalves

---

## Round 4 · List of Changes

- Added table of contents at the beginning of the manuscript;
- Corrected typos spotted by the Editor and the Referees and implemented their suggestions. In particular:
-- Added an inset to Fig.2(b) an example with results of the prediction of self-dual points through commensurate approximants having different unit cell sizes, in comparison with the estimation of the critical point through the inverse participation ratio for a large system size;
-- Added a dashed box around Figs.2(e,f) connecting them with Fig2.(d) to make it clear that the former were obtained for the model in the latter;
-- Clarified the purpose of the grey and magenta shaded regions in the caption of Fig.4;
-- Added to each figure caption reminders of the model and parameters being studied in that figure, and added hyperlinks to the relevant equations where these were defined;
-- Specified the Appendix sections that we are referring to whenever they are mentioned in the main sections, for a faster navigation throughout the manuscript.
- Added a discussion on some questions raised by in Referee report 2 to the "Discussion" section (paragraph "The scope of this work (...)");
- Added an Appendix (Appendix F).

---

## Editorial Decision

published